# Two Types of Trilocality of Probability and Correlation Tensors

**DOI:** 10.3390/e25020273

**Published:** 2023-02-01

**Authors:** Shu Xiao, Huaixin Cao, Zhihua Guo, Kanyuan Han

**Affiliations:** School of Mathematics and Statistics, Shaanxi Normal University, Xi’an 710119, China

**Keywords:** C-trilocality, D-trilocality, bell locality, probability tensor, correlation tensor

## Abstract

In this work, we discuss two types of trilocality of probability tensors (PTs) $P=〚P(a_{1}a_{2}a_{3})〛$ over an outcome set $\Omega_{3}$ and correlation tensors (CTs) $P=〚P(a_{1}a_{2}a_{3}|x_{1}x_{2}x_{3})〛$ over an outcome-input set $\Delta_{3}$ based on a triangle network and described by continuous (integral) and discrete (sum) trilocal hidden variable models (C-triLHVMs and D-triLHVMs). We say that a PT (or CT) P is C-trilocal (resp. D-trilocal) if it can be described by a C-triLHVM (resp. D-triLHVM). It is proved that a PT (resp. CT) is D-trilocal if and only if it can be realized in a triangle network by three shared separable states and a local POVM (resp. a set of local POVMs) performed at each node; a CT is C-trilocal (resp. D-trilocal) if and only if it can be written as a convex combination of the product deterministic CTs with a C-trilocal (resp. D-trilocal) PT as a coefficient tensor. Some properties of the sets consisting of C-trilocal and D-trilocal PTs (resp. C-trilocal and D-trilocal CTs) are proved, including their path-connectedness and partial star-convexity.

## 1. Introduction

Quantum networks [1,2,3,4] have recently attracted much interest as they have been identified as a promising platform for quantum information processing, such as long-distance quantum communication [5,6]. In an abstract sense, a quantum network consists of several sources, which distribute entangled quantum states to spatially separated nodes; then, the quantum information is processed locally in these nodes. This may be seen as a generalization of a classical causal model [7,8], where the shared classical information between the nodes is replaced by quantum states. Clearly, it is important to understand the quantum correlations that arise in such a quantum network. Recent developments have shown that the network structure and topology lead to novel notions of nonlocality [9,10], as well as new concepts of entanglement and separability [11,12,13], which differ from the traditional concepts and definitions [14,15]. Dealing with these new concepts requires theoretical tools for their analysis. Thus far, examples of entanglement criteria for the network scenario have been derived using the mutual information [11,12], the fidelity with pure states [12,13], or covariance matrices build from measurement probabilities [16,17]. According to Bell’s local causality assumption [18,19], the different systems measured in the experiment are considered to be all in an initial joint “hidden” state λ, where λ is arbitrary and could even describe the state of the entire universe prior to the measurement choices. The probability P(o|m,λ) of obtaining measurement outcome *o* of any particular system can depend arbitrarily on the global state λ and on the type *m* of measurement performed on that system, but not on the measurements performed on distant systems.

Focusing on quantum networks, a completely different approach to multipartite nonlocality was proposed [20,21,22]. For the case where distant observers share entanglement distributed by independent several sources, the observers may correlate distant quantum systems and establish strong correlations across the entire network by performing joint entangled measurements, such as the well-known Bell state measurement used in quantum teleportation [23]. It turns out that this situation is fundamentally different from standard multipartite nonlocality, and allows for radically novel phenomena. As regards correlations, it is now possible to witness quantum nonlocality in experiments where all the observers perform a fixed measurement; i.e., they receive no input [24,25,26,27]. This effect of quantum nonlocality without inputs is remarkable, and radically departs from previous forms of quantum nonlocality [9].

Recently, Kraft et al. [28] demonstrated that the theory of quantum coherence provides powerful tools for analyzing correlations in quantum networks and provided a direct link between the theory of multisubspace coherence [29,30] and the approach to quantum networks using covariance matrices [16,17]. Patricia et al. [31] derived sufficient conditions for entanglement to give rise to genuine multipartite nonlocality in networks and found that any network where the parties are connected by bipartite pure entangled states is genuine multipartite nonlocal, independently of the amount of entanglement in the shared states and of the topology of the network. Šupić et al. [32] introduced a notion of genuine network quantum nonlocality and showed several examples of correlations that are genuine network nonlocal, considering the so-called bilocality network of entanglement swapping. Recently, Tavakoli et al. [33] contributed a review paper by discussing the main concepts, methods, results, and future challenges in the emerging topic of Bell nonlocality in networks. Some open problems were listed at the end of their paper. In particular, the authors said that, “in the triangle network with no inputs and binary outputs, the conjecture that the local and quantum sets are identical remains open”.

When a triangle network consisting of three quantum systems $S_{1},S_{2}$ and $S_{3}$ (refer to Figure 1 below) is locally measured one time, the probabilities $P(a_{1},a_{2},a_{3})$ of obtaining outcomes $a_{1},a_{2},a_{3}$ at nodes $S_{1},S_{2}$ and $S_{3}$ form a nonnegative tensor $P=〚P(a_{1},a_{2},a_{3})〛$ over $\Omega_{3}=\left[o_{1}\right]\times \left[o_{2}\right]\times \left[o_{3}\right]$ with
$$\sum_{a_{1},a_{2},a_{3}}P\left(a_{1},a_{2},a_{3}\right)=1,$$

$\left[o_{i}\right]$ denotes the set consisting of outcomes $1,2,\ldots ,o_{i}$ at node $S_{i}$. We call it a probability tensor (PT) over $\Omega_{3}$. When a triangle network is locally measured many times, the conditional probabilities $P(a_{1}a_{2}a_{3}|x_{1}x_{2}x_{3})$ of obtaining outcomes $a_{1},a_{2},a_{3}$ at nodes $S_{1},S_{2}$ and $S_{3}$ form a nonnegative tensor $P=〚P(a_{1},a_{2},a_{3}|x_{1},x_{2},x_{3})〛$ over $\Delta_{3}=\Omega_{3}\times \left[m_{1}\right]\times \left[m_{2}\right]\times \left[m_{3}\right]$ with
$$\sum_{a_{1},a_{2},a_{3}}P\left(a_{1},a_{2},a_{3}|x_{1},x_{2},x_{3}\right)=1$$
for all $\left(x_{1},x_{2},x_{3}\right)\in \left[m_{1}\right]\times \left[m_{2}\right]\times \left[m_{3}\right]$, $\left[m_{i}\right]$ denotes the set consisting of inputs $1,2,\ldots ,m_{i}$ at node $S_{i}$. We call it a correlation tensor (CT) over $\Delta_{3}$.

In this work, we aim to introduce and discuss two types of trilocality of PTs and CTs, called C-trilocality and D-trilocality, according to their descriptions of continuous (integral) and discrete (sum) the types of trilocal hidden variable models. In Section 2, we will define and discuss the C-trilocality and D-trilocality of a PT. Section 3 is devoted to introduce and discuss the C-trilocality and D-trilocality of a CT. In Section 4, we will give a summary and list some open questions.

**Figure 1 entropy-25-00273-f001:**
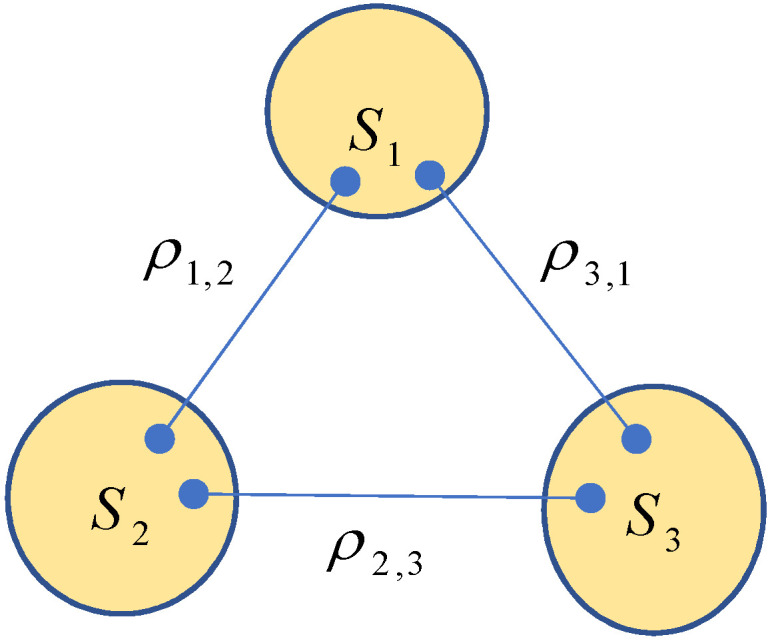
A triangle quantum network where the Hilbert spaces of systems $S_{1},S_{2}$ and $S_{3}$ are $H^{\left(1\right)}=H_{2}^{\left(1\right)}⊗H_{1}^{\left(1\right)},H^{\left(2\right)}=H_{1}^{\left(2\right)}⊗H_{2}^{\left(2\right)},$ and $H^{\left(3\right)}=H_{1}^{\left(3\right)}⊗H_{2}^{\left(3\right)},$ respectively.

## 2. Trilocality of Probability Tensors

In what follows, we use $H_{A}$ and $H_{B}$ to denote the finite-dimensional complex Hilbert spaces describing quantum systems *A* and *B*, respectively. The composite system of *A* and *B* is then described by the Hilbert space $H_{AB}:=H_{A}⊗H_{B}$. We also use $I_{X}$ and $D_{X}$ to denote the identity operator on a Hilbert space $H_{X}$ and the set of all quantum states of the system *X* described by $H_{X}$, respectively, where X=A,B and AB. We also use the notation [m]={1,2,…,m} for every positive integer *m*.

### 2.1. Triangle Quantum Networks

Considering a system-based network N with *N* nodes $S_{n}$ (quantum systems), the topological structure of the network can be described by a directed graph G(N)=(V(N),E(N)) with the set $V\left(N\right)=\left\{S_{1},S_{2},\ldots ,S_{N}\right\}$ of vertices and the set E(N) of edges where $S_{i}S_{j}\in E\left(N\right)$ if and only if $S_{i}$ and $S_{j}$ share a resource $\rho_{i,j}$ (a quantum state of a system $H_{i}⊗H_{j}$). Put $n\left(S_{i}\right)=\left\{S_{j}:S_{i}S_{j}\in E\left(N\right)\right\}$ and assume that each node shares a resource with at least one node, i.e., $n(S_{i})\neq \emptyset$ for all i=1,2,…,N. The state $\rho_{N}$ of the network N, called the network state, is the tensor product of all shared states $\rho_{i,j}$ in a certain order that you chose. Clearly, the feature of a network N is determined by its topology G(N) together with its network state $\rho_{N}$.

For example, for a triangle network TN given by Figure 1, we have
$$V\left(\mathrm{TN}\right)=\left\{S_{1},S_{2},S_{3}\right\},E\left(\mathrm{TN}\right)=\left\{S_{1}S_{2},S_{2}S_{3},S_{3}S_{1}\right\},$$
and the network state $\rho_{\mathrm{TN}}$ of TN reads
(1)$$\rho_{\mathrm{TN}}=\rho_{1,2}⊗\rho_{2,3}⊗\rho_{3,1}\in D\left(H_{1}^{\left(1\right)}⊗H_{1}^{\left(2\right)}⊗H_{2}^{\left(2\right)}⊗H_{1}^{\left(3\right)}⊗H_{2}^{\left(3\right)}⊗H_{2}^{\left(1\right)}\right),$$
where
(2)$$\rho_{1,2}\in D\left(H_{1}^{\left(1\right)}⊗H_{1}^{\left(2\right)}\right),\rho_{2,3}\in D\left(H_{2}^{\left(2\right)}⊗H_{1}^{\left(3\right)}\right),\rho_{3,1}\in D\left(H_{2}^{\left(3\right)}⊗H_{2}^{\left(1\right)}\right).$$

To explore the property of the network, a POVM measurement $M^{\left(n\right)}={\left\{M_{a_{n}}^{\left(n\right)}\right\}}_{a_{n}=1}^{d_{n}}$ is performed at each node $S_{n}$. Put $M={\left\{M^{\left(n\right)}\right\}}_{n=1}^{N}$. The observed probability distribution over the outcomes reads
(3)$$P_{N}^{M}\left(a_{1},\ldots ,a_{N}\right)=\mathrm{tr}\left[\left({⊗}_{n=1}^{N}M_{a_{n}}^{\left(n\right)}\right)\tilde{\rho_{N}}\right]$$
where ${⊗}_{n=1}^{N}M_{a_{n}}^{\left(n\right)}$ are positive operators on the Hilbert space $H_{\mathrm{net}}:={⊗}_{i=1}^{N}H^{\left(i\right)}$, $\tilde{\rho_{N}}$ denotes the state of $H_{\mathrm{net}}$ obtained from the network state $\rho_{N}$ after performing the canonical unitary transformation T from the space $H_{\mathrm{state}}$ of $\rho_{N}$ onto $H_{\mathrm{net}}$, i.e., $\tilde{\rho_{N}}=T\rho_{N}T^{†}.$ We call $\tilde{\rho_{N}}$ the measurement state.

Let us consider the triangle network given by Figure 1. To find out the state $\tilde{\rho_{\mathrm{TN}}}$, we write
$$\rho_{1,2}=\sum_{i=1}^{r}\alpha \left(i\right)X_{i}^{\left(1\right)}⊗X_{i}^{\left(2\right)}\in D\left(H_{1}^{\left(1\right)}⊗H_{1}^{\left(2\right)}\right),$$
$$\rho_{2,3}=\sum_{j=1}^{s}\beta \left(j\right)Y_{j}^{\left(2\right)}⊗Y_{j}^{\left(3\right)}\in D\left(H_{2}^{\left(2\right)}⊗H_{1}^{\left(3\right)}\right),$$
$$\rho_{3,1}=\sum_{k=1}^{t}\gamma \left(k\right)Z_{k}^{\left(3\right)}⊗Z_{k}^{\left(1\right)}\in D\left(H_{2}^{\left(3\right)}⊗H_{2}^{\left(1\right)}\right).$$
Thus, the network state reads
$$\rho_{\mathrm{TN}}=\sum_{i,j,k}\alpha \left(i\right)\beta \left(j\right)\gamma \left(k\right)\left(X_{i}^{\left(1\right)}⊗X_{i}^{\left(2\right)}\right)⊗\left(Y_{j}^{\left(2\right)}⊗Y_{j}^{\left(3\right)}\right)⊗\left(Z_{k}^{\left(3\right)}⊗Z_{k}^{\left(1\right)}\right),$$
resulting in the measurement state
$$\tilde{\rho_{\mathrm{TN}}}=\sum_{i,j,k}\alpha \left(i\right)\beta \left(j\right)\gamma \left(k\right)\left(Z_{k}^{\left(1\right)}⊗X_{i}^{\left(1\right)}\right)⊗\left(X_{i}^{\left(2\right)}⊗Y_{j}^{\left(2\right)}\right)⊗\left(Y_{j}^{\left(3\right)}⊗Z_{k}^{\left(3\right)}\right),$$
a state of
$$H^{\left(1\right)}⊗H^{\left(2\right)}⊗H^{\left(3\right)}=\left(H_{2}^{\left(1\right)}⊗H_{1}^{\left(1\right)}\right)⊗\left(H_{1}^{\left(2\right)}⊗H_{2}^{\left(2\right)}\right)⊗\left(H_{1}^{\left(3\right)}⊗H_{2}^{\left(3\right)}\right).$$
Here, the action of T is
$$|x_{1}^{\left(1\right)}x_{1}^{\left(2\right)}x_{2}^{\left(2\right)}x_{1}^{\left(3\right)}x_{2}^{\left(3\right)}x_{2}^{\left(1\right)}\rangle \mapsto |x_{2}^{\left(1\right)}x_{1}^{\left(1\right)}\rangle ⊗|x_{1}^{\left(2\right)}x_{2}^{\left(2\right)}\rangle ⊗|x_{1}^{\left(3\right)}x_{2}^{\left(3\right)}\rangle$$
for all $|x_{j}^{\left(i\right)}\rangle \in H_{j}^{\left(i\right)}.$ The joint probability is given by
(4)$$\begin{array}{ccc} P_{\mathrm{TN}}^{M}\left(a_{1},a_{2},a_{3}\right) & = & \mathrm{tr}[\left({⊗}_{n=1}^{3}M_{a_{n}}^{\left(n\right)}\right)\tilde{\rho_{\mathrm{TN}}}]\\ & = & \sum_{i,j,k}\alpha \left(i\right)\beta \left(j\right)\gamma \left(k\right)\mathrm{tr}\left[M_{a_{1}}^{\left(1\right)}\left(Z_{k}^{\left(1\right)}⊗X_{i}^{\left(1\right)}\right)\right]\\ & & \times \mathrm{tr}\left[M_{a_{2}}^{\left(2\right)}\left(X_{i}^{\left(2\right)}⊗Y_{j}^{\left(2\right)}\right)\right]\times \mathrm{tr}[M_{a_{3}}^{\left(3\right)}\left(Y_{j}^{\left(3\right)}⊗Z_{k}^{\left(3\right)})\right]. \end{array}$$

In particular, when the shared states $\rho_{i,j}$ are separable, they can be written as convex combinations of product states. Then, we can assume that the coefficients α(i),β(j),γ(k) are probability distributions (PDs) of i,j,k and that the operators $X_{i}^{\left(t\right)},Y_{j}^{\left(t\right)}$ and $Z_{K}^{\left(t\right)}$ are all states. Put
$$P_{1}\left(a_{1}|k,i\right)=\mathrm{tr}\left[M_{a_{1}}^{\left(1\right)}\left(Z_{k}^{\left(1\right)}⊗X_{i}^{\left(1\right)}\right)\right],$$
$$P_{2}\left(a_{2}|i,j\right)=\mathrm{tr}\left[M_{a_{2}}^{\left(2\right)}\left(X_{i}^{\left(2\right)}⊗Y_{j}^{\left(2\right)}\right)\right],$$
$$P_{3}\left(a_{3}|j,k\right)=\mathrm{tr}[M_{a_{3}}^{\left(3\right)}\left(Y_{j}^{\left(3\right)}⊗Z_{k}^{\left(3\right)})\right],$$
which are PDs of outcomes $a_{1},a_{2},a_{3}$, respectively. Thus, in this case, Equation (Equation 4) becomes
(5)$$\begin{array}{c} P_{\mathrm{TN}}^{M}\left(a_{1},a_{2},a_{3}\right)=\sum_{i,j,k}\alpha \left(i\right)\beta \left(j\right)\gamma \left(k\right)P_{1}\left(a_{1}|k,i\right)P_{2}\left(a_{2}|i,j\right)P_{3}\left(a_{3}|j,k\right) \end{array}$$
for all possible $a_{1},a_{2},a_{3}$. This is just the motivation for introducing the concept of D-trilocality; see Section 2.2.

### 2.2. Trilocality of Probability Tensors

The central question is whether a given probability distribution may originate from a network with a given topology [28]. The usual Bell nonlocality of a quantum state or a quantum network is the property that is exhibited by performing a set of non-compatible local POVM measurement.

Renou et al. [9] pointed out that quantum nonlocality can be demonstrated without the need of having various input settings, but only by considering the joint statistics of fixed local measurement outputs. They call this property quantum nonlocality without inputs. For example, when a triangle network is measured by just one local POVM M, joint probabilities $P_{\mathrm{TN}}^{M}\left(a_{1},a_{2},a_{3}\right)$ are obtained, which form a nonnegative tensor $P_{\mathrm{TN}}^{M}=〚P_{\mathrm{TN}}^{M}\left(a_{1},a_{2},a_{3}\right)〛$ over the index set $\Omega_{3}=\left[o_{1}\right]\times \left[o_{2}\right]\times \left[o_{3}\right]$. Generally, when a function $P:\Omega_{3}\to \left[0,1\right]$ satisfies the completeness condition:$$\sum_{a_{1},a_{2},a_{3}}P\left(a_{1},a_{2},a_{3}\right)=1,$$
we call it a probability tensor (PT) over $\Omega_{3}$, denoted by $P=〚P(a_{1},a_{2},a_{3})〛$.

Fritz in ([22] Definition 2.12) called a probability tensor $P=〚P(a_{1},a_{2},a_{3})〛$ over $\Omega_{3}$ *classical in $C_{3}$* if it can be written as
(6)$$\begin{array}{ccc} P(a_{1},a_{2},a_{3}) & = & {\;\int \int \int }_{\Lambda }q_{1}\left(\lambda_{1}\right)q_{2}\left(\lambda_{2}\right)q_{3}\left(\lambda_{3}\right)P_{1}\left(a_{1}|\lambda_{3}\lambda_{1}\right)P_{2}\left(a_{2}|\lambda_{1}\lambda_{2}\right)\\ & & \times P_{3}\left(a_{3}|\lambda_{2}\lambda_{3}\right)d\lambda_{1}d\lambda_{2}d\lambda_{3} \end{array}$$
for appropriate (conditional) distributions $q_{1}\left(\lambda_{1}\right),q_{2}\left(\lambda_{2}\right),q_{3}\left(\lambda_{3}\right),P_{1}\left(a_{1}|\lambda_{3}\lambda_{1}\right),$$P_{2}\left(a_{2}|\lambda_{1}\lambda_{2}\right),$ and $P_{3}\left(a_{3}|\lambda_{2}\lambda_{3}\right)$. It was proved ([22] Proposition 2.13) that classical correlations in $C_{3}$ are monogamous in the sense that $a_{1}$ is independent of $\lambda_{1}$ (i.e., $I(a_{1}:\lambda_{1})=0)$ and $a_{3}$ is independent of $\lambda_{2}$ (i.e., $I(a_{3}:\lambda_{2})=0)$ whenever $P(a_{1}=a_{3})=1$. Since the representation (Equation 6) is given by the integral of hidden variables, we call it a *continuous trilocal hidden variable model* (C-triLHVM) for P.

Motivated by this work, we introduce the following concepts of trilocality of tripartite PTs.

**Definition** **1.** 
*Let $P=〚P(a_{1},a_{2},a_{3})〛$ be a PT over $\Omega_{3}$.*


(1) P is said to be *C-trilocal* if it has a C-triLHVM:(7)$$\begin{array}{ccc} P(a_{1},a_{2},a_{3}) & = & {\;\int \int \int }_{\Lambda_{1}\times \Lambda_{2}\times \Lambda_{3}}q_{1}\left(\lambda_{1}\right)q_{2}\left(\lambda_{2}\right)q_{3}\left(\lambda_{3}\right)P_{1}\left(a_{1}|\lambda_{3}\lambda_{1}\right)P_{2}\left(a_{2}|\lambda_{1}\lambda_{2}\right)\\ & & \times P_{3}\left(a_{3}|\lambda_{2}\lambda_{3}\right)d\mu_{1}\left(\lambda_{1}\right)d\mu_{2}\left(\lambda_{2}\right)d\mu_{3}\left(\lambda_{3}\right) \end{array}$$
for some product measure space
$$\left(\Lambda ,\Sigma ,\mu \right)=\left(\Lambda_{1}\times \Lambda_{2}\times \Lambda_{3},\Sigma_{1}\times \Sigma_{2}\times \Sigma_{3},\mu_{1}\times \mu_{2}\times \mu_{3}\right),$$
where $\lambda =(\lambda_{1},\lambda_{2},\lambda_{3})$, $d\mu \left(\lambda \right)=d\mu_{1}\left(\lambda_{1}\right)d\mu_{2}\left(\lambda_{2}\right)d\mu_{3}\left(\lambda_{3}\right),$ and

(a) $q_{j}\left(\lambda_{j}\right)$ is a density function (DF) of $\lambda_{j}$, i.e., $q_{j}\left(\lambda_{j}\right)\geq 0$ for all $\lambda_{j}$ in $\Lambda_{j}$ such that $\int_{\Lambda_{j}}q_{j}\left(\lambda_{j}\right)d\mu_{j}\left(\lambda_{j}\right)=1$;

(b) $P_{1}\left(a_{1}|\lambda_{3}\lambda_{1}\right),P_{2}\left(a_{2}|\lambda_{1}\lambda_{2}\right)$ and $P_{3}\left(a_{3}|\lambda_{2}\lambda_{3}\right)$, called response functions (RSs) at nodes 1,2 and 3, are PDs of $a_{1},a_{2}$ and $a_{3}$, respectively, for each $\lambda =(\lambda_{1},\lambda_{2},\lambda_{3})$ in Λ and are Ω-measurable on Λ w.r.t. $\lambda =(\lambda_{1},\lambda_{2},\lambda_{3})$ for each $a=(a_{1},a_{2},a_{3})$ in $\Omega_{3}$.

(2) P is said to be *D-trilocal* if it has a D-triLHVM:(8)$$\begin{array}{c} P\left(a_{1},a_{2},a_{3}\right)=\sum_{\lambda_{1}=1}^{n_{1}}\sum_{\lambda_{2}=1}^{n_{2}}\sum_{\lambda_{3}=1}^{n_{3}}q_{1}\left(\lambda_{1}\right)q_{2}\left(\lambda_{2}\right)q_{3}\left(\lambda_{3}\right)P_{1}\left(a_{1}|\lambda_{3}\lambda_{1}\right)P_{2}\left(a_{2}|\lambda_{1}\lambda_{2}\right)P_{3}\left(a_{3}|\lambda_{2}\lambda_{3}\right) \end{array}$$
for all $a_{k}\in \left[o_{k}\right]\left(k=1,2,3\right)$, where $q_{k}\left(\lambda_{k}\right)$, $P_{1}\left(a_{1}|\lambda_{3}\lambda_{1}\right),P_{2}\left(a_{2}|\lambda_{1}\lambda_{2}\right)$ and $P_{3}\left(a_{3}|\lambda_{2}\lambda_{3}\right)$ are PDs of $\lambda_{k}$, $a_{1},a_{2}$ and $a_{3}$, respectively.

(3) P is said to be *C-nontrilocal* (resp. *D-nontrilocal*) if it is not C-trilocal (resp. not D-trilocal).

Please refer to Figure 2.

We use ${\mathrm{PT}}^{C-\mathrm{trilocal}}\left(\Omega_{3}\right)$ and ${\mathrm{PT}}^{D-\mathrm{trilocal}}\left(\Omega_{3}\right)$ to denote the sets of all C-trilocal and D-trilocal PTs over $\Omega_{3}$, respectively. Obviously, ${\mathrm{PT}}^{C-\mathrm{trilocal}}\left(\Omega_{3}\right)⊃{\mathrm{PT}}^{D-\mathrm{trilocal}}\left(\Omega_{3}\right).$

When P has a C-triLHVM (Equation 7), by letting
$$d\gamma_{k}\left(\lambda_{k}\right)=q_{k}\left(\lambda_{k}\right)d\mu_{k}\left(\lambda_{k}\right)\left(k=1,2,3\right),$$
equivalently, defining measures $\gamma_{k}$ on $\Sigma_{k}$ as
$$\gamma_{k}\left(E_{k}\right)=\int_{\Lambda_{k}}\chi_{E_{k}}\left(\lambda_{k}\right)q_{k}\left(\lambda_{k}\right)d\mu_{k}\left(\lambda_{k}\right),\;\forall E_{k}\in \Sigma_{k},$$
where $\chi_{E_{k}}\left(\lambda_{k}\right)$ is the characteristic function of $E_{k}$, we obtain a product probability space
$$\left(\Lambda ,\Sigma ,\gamma \right)=\left(\Lambda_{1}\times \Lambda_{2}\times \Lambda_{3},\Sigma_{1}\times \Sigma_{2}\times \Sigma_{3},\gamma_{1}\times \gamma_{2}\times \gamma_{3}\right).$$
In this setting, the C-triLHVM (Equation 7) becomes
(9)$$\begin{array}{c} P\left(a_{1},a_{2},a_{3}\right)=\int_{\Lambda }P_{1}\left(a_{1}|\lambda_{3}\lambda_{1}\right)P_{2}\left(a_{2}|\lambda_{1}\lambda_{2}\right)P_{3}\left(a_{3}|\lambda_{2}\lambda_{3}\right)d\gamma \left(\lambda \right), \end{array}$$
where $d\gamma \left(\lambda \right)=d\gamma_{1}\left(\lambda_{1}\right)d\gamma_{2}\left(\lambda_{2}\right)d\gamma_{3}\left(\lambda_{3}\right).$

Conversely, every C-triLHVM (Equation 9) can be written as a C-triLHVM (Equation 7) by letting $q_{k}\left(\lambda_{k}\right)\equiv 1$.

This leads to the following conclusion.

**Proposition** **1.** 
*A tripartite PT $P=〚P(a_{1},a_{2},a_{3})〛$ over $\Omega_{3}$ is C-trilocal if and only if it admits a C-triLHVM (Equation 9) for a product probability space*

(Λ,Σ,γ)=(Λ1×Λ2×Λ3,Σ1×Σ2×Σ3,γ1×γ2×γ3).



**Example** **1.** 
*Consider the PT $P_{\mathrm{cube}}=〚P\left(a_{1},a_{2},a_{3}\right)〛$ over $\Omega_{3}$ defined by Riemann integral*

(10)
$$\begin{array}{c} P\left(a_{1},a_{2},a_{3}\right)={\;\int \int \int }_{{\left[0,1\right]}^{3}}P_{1}\left(a_{1}|\lambda_{3}\lambda_{1}\right)P_{2}\left(a_{2}|\lambda_{1}\lambda_{2}\right)P_{3}\left(a_{3}|\lambda_{2}\lambda_{3}\right)d\lambda_{1}d\lambda_{2}d\lambda_{3}, \end{array}$$

*where*

$$P_{1}\left(a_{1}|\lambda_{3}\lambda_{1}\right)=\frac{\cos(a_{1}\lambda_{3}\lambda_{1}/o_{1})}{\sum_{k_{1}=1}^{o_{1}}\cos\left(k_{1}\lambda_{3}\lambda_{1}/o_{1}\right)},$$


$$P_{2}\left(a_{2}|\lambda_{1}\lambda_{2}\right)=\frac{\cos(a_{2}\lambda_{1}\lambda_{2}/o_{2})}{\sum_{k_{2}=1}^{o_{2}}\cos\left(k_{2}\lambda_{1}\lambda_{2}/o_{2}\right)},$$


$$P_{3}\left(a_{3}|\lambda_{2}\lambda_{3}\right)=\frac{\cos(a_{3}\lambda_{2}\lambda_{3}/o_{3})}{\sum_{k_{3}=1}^{o_{3}}\cos\left(k_{3}\lambda_{2}\lambda_{3}/o_{3}\right)},$$

*which are PDs of $a_{1},a_{2},a_{3}$, respectively, and measurable w.r.t. Lebesgue measure $\left(\gamma_{1},\gamma_{2},\gamma_{3}\right)$ on $\Lambda ={\left[0,1\right]}^{3}$. $P_{\mathrm{cube}}$ is clearly a C-trilocal PT over $\Omega_{3}$ using Proposition 1.*


Moreover, if we replace the space $\Lambda ={\left[0,1\right]}^{3}$ of hidden variables in Example 1 with $\Lambda ={\left[-1,1\right]}^{3}$ and take $p_{i}\left(\lambda_{i}\right)=\frac{1}{2}$ for i=1,2,3, then the PT $P=〚P(a_{1},a_{2},a_{3})〛$ defined by
(11)$$\begin{array}{c} P\left(a_{1},a_{2},a_{3}\right)={\;\int \int \int }_{{\left[-1,1\right]}^{3}}p_{1}\left(\lambda_{1}\right)p_{2}\left(\lambda_{2}\right)p_{3}\left(\lambda_{3}\right)P_{1}\left(a_{1}|\lambda_{3}\lambda_{1}\right)P_{2}\left(a_{2}|\lambda_{1}\lambda_{2}\right)P_{3}\left(a_{3}|\lambda_{2}\lambda_{3}\right)d\lambda_{1}d\lambda_{2}d\lambda_{3} \end{array}$$
is also C-trilocal.

**Question 1.** Consider the PT $P_{\mathrm{ball}}=〚P\left(a_{1},a_{2},a_{3}\right)〛$ over $\Omega_{3}$ given by Riemann integral
(12)$$\begin{array}{c} P\left(a_{1},a_{2},a_{3}\right)=\frac{3}{4\pi }{\;\int \int \int }_{\Lambda }P_{1}\left(a_{1}|\lambda_{3}\lambda_{1}\right)P_{2}\left(a_{2}|\lambda_{1}\lambda_{2}\right)P_{3}\left(a_{3}|\lambda_{2}\lambda_{3}\right)d\lambda_{1}d\lambda_{2}d\lambda_{3}, \end{array}$$
where Λ denotes the closed unit ball in $R^{3}$ and the PDs $P_{1}\left(a_{1}|\lambda_{3}\lambda_{1}\right),P_{2}\left(a_{2}|\lambda_{1}\lambda_{2}\right)$ and $P_{3}\left(a_{3}|\lambda_{2}\lambda_{3}\right)$ are as in Example 1. An interesting question is whether $P_{\mathrm{ball}}$ is C-trilocal.

It is remarkable to mention that a C-triLHVM for a PT must be given by an integral that is taken over a *product space* $\Lambda_{1}\times \Lambda_{2}\times \Lambda_{3}$ due to the independence of the hidden variables $\lambda_{1},\lambda_{2}$ and $\lambda_{3}$. It is also noted that the integrand must be a product of the three DFs of $\lambda_{1},\lambda_{2}$ and $\lambda_{3}$ and the three PDs of $a_{1},a_{2}$ and $a_{3}$ with parameters $\left(\lambda_{3},\lambda_{1}\right),\left(\lambda_{1},\lambda_{2}\right)$ and $\left(\lambda_{2},\lambda_{3}\right)$, respectively. Although the unit ball Λ in Question 1 is homeomorphic to the unit cube ${\left[0,1\right]}^{3}$ or ${\left[-1,1\right]}^{3}$, the integrand may be changed as the one that is not of the desired form. Thus, the answer to Question 1 may be very hard.

**Definition** **2.** 
*A tripartite PT $P=〚P(a_{1},a_{2},a_{3})〛$ over $\Omega_{3}$ is said to be tri-quantum if there exists a TN with the state $\rho_{\mathrm{TN}}$ and a local POVM $M=M^{\left(1\right)}⊗M^{\left(2\right)}⊗M^{\left(3\right)}$ such that $P=P_{\mathrm{TN}}^{M}$, i.e.,*

(13)
$$\begin{array}{c} P\left(a_{1},a_{2},a_{3}\right)=P_{\mathrm{TN}}^{M}\left(a_{1},a_{2},a_{3}\right):=\mathrm{tr}\left[\left({⊗}_{n=1}^{3}M_{a_{n}}^{\left(n\right)}\right)\tilde{\rho_{\mathrm{TN}}}\right],\;\forall a_{k}\in \left[o_{k}\right]. \end{array}$$

*In particular, when the shares’ states $\rho_{i,j}$ can be chosen as separable states, we say that P is separable tri-quantum.*


**Definition** **3.** 
*A triangle network TN given by Figure 1 is said to be C-trilocal (resp. D-trilocal) if, for every local POVM $M=M^{\left(1\right)}⊗M^{\left(2\right)}⊗M^{\left(3\right)}$, where $M^{\left(k\right)}={\left\{M_{a_{k}}^{\left(k\right)}\right\}}_{a_{k}=1}^{d_{k}}$, the generated PT $P_{\mathrm{TN}}^{M}=〚P_{\mathrm{TN}}\left(a_{1},a_{2},a_{3}\right)〛$ is C-trilocal (resp. D-trilocal). It is said to be non C-trilocal (resp. non D-trilocal) if it is not C-trilocal (resp. non D-trilocal), i.e., there exists an $M={\left\{M^{\left(k\right)}\right\}}_{k=1}^{3}$ such that PT $P_{\mathrm{TN}}^{M}$ is non-C-trilocal (resp. non-D-trilocal), referring to Figure 3.*


**Proposition** **2.** 
*Every separable (i.e., all shared states $\rho_{i,j}$ are separable) triangle network TN given by Figure 1 is D-trilocal.*


**Proof.** Suppose that the TN given by Figure 1 is separable. Then, the shared states $\rho_{s,t}$ are separable, i.e., there exist scalars $x_{\lambda_{1}},y_{\lambda_{2}},z_{\lambda_{3}}\in \left[0,1\right]$ satisfying
$$\sum_{\lambda_{1}=1}^{n_{1}}x_{\lambda_{1}}=1,\sum_{\lambda_{2}=1}^{n_{2}}y_{\lambda_{2}}=1,\sum_{\lambda_{3}=1}^{n_{3}}z_{\lambda_{3}}=1,$$
such that
$$\rho_{1,2}=\sum_{\lambda_{1}=1}^{n_{1}}x_{\lambda_{1}}\rho_{1}^{\left(1\right)}\left(\lambda_{1}\right)⊗\rho_{1}^{\left(2\right)}\left(\lambda_{1}\right)\in D\left(H_{1}^{\left(1\right)}⊗H_{1}^{\left(2\right)}\right),$$
$$\rho_{2,3}=\sum_{\lambda_{2}=1}^{n_{2}}y_{\lambda_{2}}\rho_{2}^{\left(2\right)}\left(\lambda_{2}\right)⊗\rho_{1}^{\left(3\right)}\left(\lambda_{2}\right)\in D\left(H_{2}^{\left(2\right)}⊗H_{1}^{\left(3\right)}\right),$$
$$\rho_{3,1}=\sum_{\lambda_{3}=1}^{n_{3}}z_{\lambda_{3}}\rho_{2}^{\left(3\right)}\left(\lambda_{3}\right)⊗\rho_{2}^{\left(1\right)}\left(\lambda_{3}\right)\in D\left(H_{2}^{\left(3\right)}⊗H_{2}^{\left(1\right)}\right),$$
where $\rho_{t}^{\left(s\right)}\left(r\right)\in D\left(H_{t}^{\left(s\right)}\right).$ Thus, the network state reads
$$\rho_{\mathrm{TN}}=\sum_{\lambda_{1},\lambda_{2},\lambda_{3}}x_{\lambda_{1}}y_{\lambda_{2}}z_{\lambda_{3}}\rho_{1}^{\left(1\right)}\left(\lambda_{1}\right)⊗\rho_{1}^{\left(2\right)}\left(\lambda_{1}\right)⊗\rho_{2}^{\left(2\right)}\left(\lambda_{2}\right)⊗\rho_{1}^{\left(3\right)}\left(\lambda_{2}\right)⊗\rho_{2}^{\left(3\right)}\left(\lambda_{3}\right)⊗\rho_{2}^{\left(1\right)}\left(\lambda_{3}\right),$$
which is a state of system $H_{1}^{\left(1\right)}⊗H_{1}^{\left(2\right)}⊗H_{2}^{\left(2\right)}⊗H_{1}^{\left(3\right)}⊗H_{2}^{\left(3\right)}⊗H_{2}^{\left(1\right)}$, and then the measurement state is
$$\tilde{\rho_{\mathrm{TN}}}=\sum_{\lambda_{1},\lambda_{2},\lambda_{3}}x_{\lambda_{1}}y_{\lambda_{2}}z_{\lambda_{3}}\left(\rho_{2}^{\left(1\right)}\left(\lambda_{3}\right)⊗\rho_{1}^{\left(1\right)}\left(\lambda_{1}\right)\right)⊗\left(\rho_{1}^{\left(2\right)}\left(\lambda_{1}\right)⊗\rho_{2}^{\left(2\right)}\left(\lambda_{2}\right)\right)⊗\left(\rho_{1}^{\left(3\right)}\left(\lambda_{2}\right)⊗\rho_{2}^{\left(3\right)}\left(\lambda_{3}\right)\right),$$
being a state of system
$$H^{\left(1\right)}⊗H^{\left(2\right)}⊗H^{\left(3\right)}=\left(H_{2}^{\left(1\right)}⊗H_{1}^{\left(1\right)}\right)⊗\left(H_{1}^{\left(2\right)}⊗H_{2}^{\left(2\right)}\right)⊗\left(H_{1}^{\left(3\right)}⊗H_{2}^{\left(3\right)}\right).$$
For every local POVM measurement, $M=M^{\left(1\right)}⊗M^{\left(2\right)}⊗M^{\left(3\right)}$ of system $H^{\left(1\right)}⊗H^{\left(2\right)}⊗H^{\left(3\right)}$, where $M^{\left(k\right)}={\left\{M_{a_{k}}^{\left(k\right)}\right\}}_{a_{k}=1}^{d_{k}}$, we have
$$\begin{array}{ccc} P_{\mathrm{TN}}^{M}\left(a_{1},a_{2},a_{3}\right) & = & \mathrm{tr}[\left({⊗}_{n=1}^{3}M_{a_{n}}^{\left(n\right)}\right)\tilde{\rho_{\mathrm{TN}}}]\\ & = & \sum_{\lambda_{1},\lambda_{2},\lambda_{3}}x_{\lambda_{1}}y_{\lambda_{2}}z_{\lambda_{3}}\mathrm{tr}\left[M_{a_{1}}^{\left(1\right)}\left(\rho_{1}^{\left(1\right)}\left(\lambda_{1}\right)⊗\rho_{2}^{\left(1\right)}\left(\lambda_{3}\right)\right)\right]\\ & & \times \mathrm{tr}\left[M_{a_{2}}^{\left(2\right)}\left(\rho_{1}^{\left(2\right)}\left(\lambda_{1}\right)⊗\rho_{2}^{\left(2\right)}\left(\lambda_{2}\right)\right)\right]\mathrm{tr}\left[M_{a_{3}}^{\left(3\right)}\left(\rho_{1}^{\left(3\right)}\left(\lambda_{2}\right)⊗\rho_{2}^{\left(3\right)}\left(\lambda_{3}\right)\right)\right]\\ & = & \sum_{\lambda_{1},\lambda_{2},\lambda_{3}}q_{1}\left(\lambda_{1}\right)q_{2}\left(\lambda_{2}\right)q_{3}\left(\lambda_{3}\right)P_{1}\left(a_{1}|\lambda_{3}\lambda_{1}\right)P_{2}\left(a_{2}|\lambda_{1}\lambda_{2}\right)P_{3}\left(a_{3}|\lambda_{2}\lambda_{3}\right), \end{array}$$
for all $a_{k}\in \left[o_{k}\right]$, where $q_{1}\left(\lambda_{1}\right)=x_{\lambda_{1}},q_{2}\left(\lambda_{2}\right)=y_{\lambda_{2}},q_{3}\left(\lambda_{3}\right)=z_{\lambda_{3}}$ and
$$P_{1}\left(a_{1}|\lambda_{3}\lambda_{1}\right)=\mathrm{tr}\left[M_{a_{1}}^{\left(1\right)}\left(\rho_{1}^{\left(1\right)}\left(\lambda_{1}\right)⊗\rho_{2}^{\left(1\right)}\left(\lambda_{3}\right)\right)\right],$$
$$P_{2}\left(a_{2}|\lambda_{1}\lambda_{2}\right)=\mathrm{tr}\left[M_{a_{2}}^{\left(2\right)}\left(\rho_{1}^{\left(2\right)}\left(\lambda_{1}\right)⊗\rho_{2}^{\left(2\right)}\left(\lambda_{2}\right)\right)\right],$$
$$P_{3}\left(a_{3}|\lambda_{2}\lambda_{3}\right)=\mathrm{tr}\left[M_{a_{3}}^{\left(3\right)}\left(\rho_{1}^{\left(3\right)}\left(\lambda_{2}\right)⊗\rho_{2}^{\left(3\right)}\left(\lambda_{3}\right)\right)\right].$$
Clearly,
$${\left\{q_{k}\left(\lambda_{k}\right)\right\}}_{\lambda_{k}\in \left[n_{k}\right]},{\left\{P_{1}\left(a_{1}|\lambda_{3}\lambda_{1}\right)\right\}}_{a_{1}\in \left[o_{1}\right]},{\left\{P_{2}\left(a_{2}|\lambda_{1}\lambda_{2}\right)\right\}}_{a_{2}\in \left[o_{2}\right]},{\left\{P_{3}\left(a_{3}|\lambda_{2}\lambda_{3}\right)\right\}}_{a_{3}\in \left[o_{3}\right]}$$
are PDs. It follows from Definition 3 that the triangle network TN given by Figure 1 is D-trilocal. The proof is completed. □

**Proposition** **3.** 
*A PT P over $\Omega_{3}$ is D-trilocal if and only if it is separable tri-quantum.*


**Proof.** The sufficiency is given by Proposition 2. To show the necessity, we let $P=\{P\left(a_{1},a_{2},a_{3}\right)\}$ be a D-trilocal PT over $\Omega_{3}$. Then, it can be written as (Equation 8). Choose Hilbert spaces
$$H_{1}^{\left(1\right)}=H_{1}^{\left(2\right)}=C^{n_{1}},H_{2}^{\left(2\right)}=H_{1}^{\left(3\right)}=C^{n_{2}},H_{2}^{\left(1\right)}=H_{2}^{\left(3\right)}=C^{n_{3}},$$
take their orthonormal bases $\{|\lambda_{3}\rangle {\}}_{\lambda_{3}=1}^{n_{3}}$, $\{|\lambda_{1}\rangle {\}}_{\lambda_{1}=1}^{n_{1}}$ and $\{|\lambda_{2}\rangle {\}}_{\lambda_{2}=1}^{n_{2}}$, respectively, and put
$$H^{\left(1\right)}=H_{2}^{\left(1\right)}⊗H_{1}^{\left(1\right)}=C^{n_{3}}⊗C^{n_{1}},H^{\left(2\right)}=H_{1}^{\left(2\right)}⊗H_{2}^{\left(2\right)}=C^{n_{1}}⊗C^{n_{2}},H^{\left(3\right)}=H_{1}^{\left(3\right)}⊗H_{2}^{\left(3\right)}=C^{n_{2}}⊗C^{n_{3}}$$
and choose separable states
$$\rho_{1,2}=\sum_{\lambda_{1}=1}^{n_{1}}q_{1}\left(\lambda_{1}\right)|\lambda_{1}\rangle \langle \lambda_{1}\left|⊗\right|\lambda_{1}\rangle \langle \lambda_{1}|\in D\left(H_{1}^{\left(1\right)}⊗H_{1}^{\left(2\right)}\right)=D\left(C^{n_{1}}⊗C^{n_{1}}\right),$$
$$\rho_{2,3}=\sum_{\lambda_{2}=1}^{n_{2}}q_{2}\left(\lambda_{2}\right)|\lambda_{2}\rangle \langle \lambda_{2}\left|⊗\right|\lambda_{2}\rangle \langle \lambda_{2}|\in D\left(H_{2}^{\left(2\right)}⊗H_{1}^{\left(3\right)}\right)=D\left(C^{n_{2}}⊗C^{n_{2}}\right),$$
$$\rho_{3,1}=\sum_{\lambda_{3}=1}^{n_{3}}q_{3}\left(\lambda_{3}\right)|\lambda_{3}\rangle \langle \lambda_{3}\left|⊗\right|\lambda_{3}\rangle \langle \lambda_{3}|\in D\left(H_{2}^{\left(3\right)}⊗H_{2}^{\left(1\right)}\right)=D\left(C^{n_{3}}⊗C^{n_{3}}\right),$$
then we obtain a triangle network TN with the network state
$$\begin{array}{ccc} \rho_{\mathrm{TN}} & = & \rho_{1,2}⊗\rho_{2,3}⊗\rho_{3,1}\\ & = & \sum_{\lambda_{1},\lambda_{2},\lambda_{3}}q_{1}\left(\lambda_{1}\right)q_{2}\left(\lambda_{2}\right)q_{3}\left(\lambda_{3}\right)\\ & & \times |\lambda_{1}\rangle \langle \lambda_{1}\left|⊗\right|\lambda_{1}\rangle \langle \lambda_{1}\left|⊗\right|\lambda_{2}\rangle \langle \lambda_{2}\left|⊗\right|\lambda_{2}\rangle \langle \lambda_{2}\left|⊗\right|\lambda_{3}\rangle \langle \lambda_{3}\left|⊗\right|\lambda_{3}\rangle \langle \lambda_{3}|, \end{array}$$
inducing the measurement state
$$\begin{array}{ccc} \tilde{\rho_{\mathrm{TN}}} & = & \sum_{\lambda_{1},\lambda_{2},\lambda_{3}}q_{1}\left(\lambda_{1}\right)q_{2}\left(\lambda_{2}\right)q_{3}\left(\lambda_{3}\right)\\ & & \times (|\lambda_{3}\rangle \langle \lambda_{3}\left|⊗\right|\lambda_{1}\rangle \langle \lambda_{1}|)⊗(|\lambda_{1}\rangle \langle \lambda_{1}\left|⊗\right|\lambda_{2}\rangle \langle \lambda_{2}|)⊗(|\lambda_{2}\rangle \langle \lambda_{2}\left|⊗\right|\lambda_{3}\rangle \langle \lambda_{3}|), \end{array}$$
in $D(H^{\left(1\right)}⊗H^{\left(2\right)}⊗H^{\left(3\right)})$. By defining separable positive operators:
$$M_{a_{1}}^{\left(1\right)}=\sum_{\lambda_{3}^{\prime }=1}^{n_{3}}\sum_{\lambda_{1}^{\prime }=1}^{n_{1}}P_{1}\left(a_{1}|\lambda_{3}^{\prime }\lambda_{1}^{\prime }\right)|\lambda_{3}^{\prime }\lambda_{1}^{\prime }\rangle \langle \lambda_{3}^{\prime }\lambda_{1}^{\prime }|,$$
$$M_{a_{2}}^{\left(2\right)}=\sum_{\lambda_{1}^{\prime }=1}^{n_{1}}\sum_{\lambda_{2}^{\prime }=1}^{n_{2}}P_{2}\left(a_{2}|\lambda_{1}^{\prime }\lambda_{2}^{\prime }\right)|\lambda_{1}^{\prime }\lambda_{2}^{\prime }\rangle \langle \lambda_{1}^{\prime }\lambda_{2}^{\prime }|,$$
$$M_{a_{3}}^{\left(3\right)}=\sum_{\lambda_{2}^{\prime }=1}^{n_{2}}\sum_{\lambda_{3}^{\prime }=1}^{n_{3}}P_{3}\left(a_{3}|\lambda_{2}^{\prime }\lambda_{3}^{\prime }\right)|\lambda_{2}^{\prime }\lambda_{3}^{\prime }\rangle \langle \lambda_{2}^{\prime }\lambda_{3}^{\prime }|$$
on Hilbert spaces $H^{\left(1\right)},H^{\left(2\right)}$ and $H^{\left(3\right)}$, respectively, we obtain POVMs ${\left\{M_{a_{k}}^{\left(k\right)}\right\}}_{a_{k}=1}^{o_{k}}$ of system $H^{\left(k\right)}$ for each k=1,2,3. Using (Equation 8) yields that
$$P\left(a_{1},a_{2},a_{3}\right)=\mathrm{tr}\left[\left({⊗}_{n=1}^{3}M_{a_{n}}^{\left(n\right)}\right)\tilde{\rho_{\mathrm{TN}}}\right],\;\forall a_{k}\in \left[o_{k}\right].$$
This shows that P is separable tri-quantum. The proof is completed. □

Recently, Tavakoli et al. [33] said that, “in the triangle network with no inputs and binary outputs, the conjecture that the local and quantum sets are identical remains open”. Proposition 3 above shows that D-trilocality and separable tri-quantum of a tripartite PT are equivalent. Renou et al. ([9] Theorem I) found a PT (they called a quantum distribution) $P_{Q}\left(a,b,c\right)$ that cannot be reproduced by any classical trilocal model (Equation 9) with deterministic response functions (DRFs) $P_{1}\left(a_{1}|\lambda_{3}\lambda_{1}\right),P_{2}\left(a_{2}|\lambda_{1}\lambda_{2}\right),P_{3}\left(a_{3}|\lambda_{2}\lambda_{3}\right)$. After a careful reading of their proof, we find that the proof of $X_{0}\cap X_{1}=\emptyset$ (for example) works well only for a D-triLHVM with DRFs. In fact, they proved that the $P_{Q}\left(a,b,c\right)$ cannot be reproduced by any D-triLHVM with DRFs. The following proposition shows that a D-triLHVM (Equation 8) can be assumed to be deterministic, i.e., the response functions are {0,1}-valued. Thus, combining ([9] Theorem I), we see that the quantum distribution $P_{Q}\left(a,b,c\right)$ is not D-trilocal. This shows that a tri-quantum PT is not necessarily D-trilocal. Thus, an interesting question is whether the $P_{Q}\left(a,b,c\right)$ is C-trilocal.

**Proposition** **4.** 
*A tripartite PT $P=〚P(a_{1},a_{2},a_{3})〛$ over $\Omega_{3}$ is D-trilocal if and only if it can be written as*

(14)
$$\begin{array}{c} P\left(a_{1},a_{2},a_{3}\right)=\sum_{\mu_{1},\mu_{2},\mu_{3}}\pi_{1}\left(\mu_{1}\right)\pi_{2}\left(\mu_{2}\right)\pi_{3}\left(\mu_{3}\right)P_{1}\left(a_{1}|\mu_{3}\mu_{1}\right)P_{2}\left(a_{2}|\mu_{1}\mu_{2}\right)P_{3}\left(a_{3}|\mu_{2}\mu_{3}\right) \end{array}$$

*for all $a_{k}\in \left[o_{k}\right]$, where ${\left\{\pi_{k}\left(\mu_{k}\right)\right\}}_{\mu_{k}\in D_{k}}$ are PDs and*

$${\left\{P_{1}\left(a_{1}|\mu_{3}\mu_{1}\right)\right\}}_{a_{1}\in \left[o_{1}\right]},{\left\{P_{2}\left(a_{2}|\mu_{1}\mu_{2}\right)\right\}}_{a_{2}\in \left[o_{2}\right]},{\left\{P_{3}\left(a_{3}|\mu_{2}\mu_{3}\right)\right\}}_{a_{3}\in \left[o_{3}\right]}$$

*are {0,1}-PDs for all $\mu_{k}$.*


**Proof.** The sufficiency is clear. To show the necessity, we assume that P is D-trilocal. Then, it can be written as (Equation 8). Since matrices
$$\left[P_{1}\left(a_{1}|\lambda_{3}\lambda_{1}\right)\right]\in R^{n_{3}n_{1}\times o_{1}},\left[P_{2}\left(a_{2}|\lambda_{1}\lambda_{2}\right)\right]\in R^{n_{1}n_{2}\times o_{2}}\;\mathrm{and}\;\left[P_{3}\left(a_{3}|\lambda_{2}\lambda_{3}\right)\right]\in R^{n_{3}n_{1}\times o_{3}}$$
are row-stochastic (RS), they can be represented as convex combinations of all {0,1}-RS matrices [34], i.e.,
$$P_{1}\left(a_{1}|\lambda_{3}\lambda_{1}\right)=\sum_{i=1}^{N_{1}}r_{i}\delta_{a_{1},J_{i}\left(\lambda_{3},\lambda_{1}\right)},P_{2}\left(a_{2}|\lambda_{1}\lambda_{2}\right)=\sum_{j=1}^{N_{2}}s_{j}\delta_{a_{2},K_{j}\left(\lambda_{1},\lambda_{2}\right)},P_{3}\left(a_{3}|\lambda_{2}\lambda_{3}\right)=\sum_{k=1}^{N_{3}}t_{k}\delta_{a_{3},L_{k}\left(\lambda_{2},\lambda_{3}\right)},$$
where $N_{1}={\left(o_{1}\right)}^{n_{3}n_{1}},N_{2}={\left(o_{2}\right)}^{n_{1}n_{2}},N_{3}={\left(o_{3}\right)}^{n_{2}n_{3}},$ and ${\left\{J_{i}\right\}}_{i=1}^{N_{1}},{\left\{K_{j}\right\}}_{j=1}^{N_{2}}$ and ${\left\{L_{k}\right\}}_{k=1}^{N_{3}}$ are the sets of all maps from $\left[n_{3}n_{1}\right]$ into $\left[o_{1}\right]$, $\left[n_{1}n_{2}\right]$ into $\left[o_{2}\right]$, and $\left[n_{2}n_{3}\right]$ into $\left[o_{3}\right]$, respectively. Using (Equation 8) yields that
$$\begin{array}{ccc} P(a_{1},a_{2},a_{3}) & = & \sum_{i,j,k}\sum_{\lambda_{1},\lambda_{2},\lambda_{3}}q_{1}\left(\lambda_{1}\right)q_{2}\left(\lambda_{2}\right)q_{3}\left(\lambda_{3}\right)r_{i}s_{j}t_{k}\delta_{a_{1},J_{i}\left(\lambda_{3},\lambda_{1}\right)}\delta_{a_{2},K_{j}\left(\lambda_{1},\lambda_{2}\right)}\delta_{a_{3},L_{k}\left(\lambda_{2},\lambda_{3}\right)}\\ & = & \sum_{\mu_{k}\in D_{k}}\pi_{1}\left(\mu_{1}\right)\pi_{2}\left(\mu_{2}\right)\pi_{3}\left(\mu_{3}\right)P_{1}\left(a_{1}|\mu_{3}\mu_{1}\right)P_{2}\left(a_{2}|\mu_{1}\mu_{2}\right)P_{3}\left(a_{3}|\mu_{2}\mu_{3}\right), \end{array}$$
where $D_{1}=\left[N_{2}\right]\times \left[n_{1}\right],D_{2}=\left[N_{3}\right]\times \left[n_{2}\right],D_{3}=\left[N_{1}\right]\times \left[n_{3}\right],$ and
$$\mu_{1}=\left(s_{j},\lambda_{1}\right),\mu_{2}=\left(t_{k},\lambda_{2}\right),\mu_{3}=\left(r_{i},\lambda_{3}\right),$$
$$\pi_{1}\left(\mu_{1}\right)=q_{1}\left(\lambda_{1}\right)s_{j},\pi_{2}\left(\mu_{2}\right)=q_{2}\left(\lambda_{2}\right)t_{k},\pi_{3}\left(\mu_{3}\right)=q_{3}\left(\lambda_{3}\right)r_{i},$$
$$P_{1}\left(a_{1}|\mu_{3}\mu_{1}\right)=\delta_{a_{1},J_{i}\left(\lambda_{3},\lambda_{1}\right)},\;P_{2}\left(a_{2}|\mu_{1}\mu_{2}\right)=\delta_{a_{2},K_{j}\left(\lambda_{1},\lambda_{2}\right)},\;P_{3}\left(a_{3}|\mu_{2}\mu_{3}\right)=\delta_{a_{3},L_{k}\left(\lambda_{2},\lambda_{3}\right)}.$$
Clearly, ${\left\{\pi_{k}\left(\mu_{k}\right)\right\}}_{\mu_{k}\in D_{k}}\left(k=1,2,3\right)$ are PDs and for all $\mu_{k}$,
$${\left\{P_{1}\left(a_{1}|\mu_{3}\mu_{1}\right)\right\}}_{a_{1}\in \left[o_{1}\right]},{\left\{P_{2}\left(a_{2}|\mu_{1}\mu_{2}\right)\right\}}_{a_{2}\in \left[o_{2}\right]},{\left\{P_{3}\left(a_{3}|\mu_{2}\mu_{3}\right)\right\}}_{a_{3}\in \left[o_{3}\right]}$$
are {0,1}-PDs. Equation (Equation 14) follows, and the proof is completed. □

To discuss geometric and topological properties of C-trilocal and D-trilocal PTs, we have to put them into a topological space. A natural way is to consider the real Hilbert space $P(\Omega_{3})$ consisting of all tensors $P=〚P(a_{1},a_{2},a_{3})〛$ over $\Delta_{3}$ defined by functions $P:\Omega_{3}\to R$, in which the operations and inner products are given by
$$sP+tQ=〚sP\left(a_{1},a_{2},a_{3}\right)+tQ\left(a_{1},a_{2},a_{3}\right)〛,\langle P|Q\rangle =\sum_{a_{i}}P\left(a_{1},a_{2},a_{3}\right)Q\left(a_{1},a_{2},a_{3}\right)$$
for all s,t∈R and all elements P and Q of $P(\Delta_{3})$. The norm induced by the inner product reads
$$∥P∥={\left(\sum_{a_{i}}{\left|P\left(a_{1},a_{2},a_{3}\right)\right|}^{2}\right)}^{\frac{1}{2}}$$
and then a sequence ${\left\{P_{n}\right\}}_{n=1}^{\infty }=\{〚P_{n}(a_{1},a_{2},a_{3})〛\}{\;}_{n=1}^{\infty }$ is convergent (in norm) to $P=〚P(a_{1},a_{2},a_{3})〛$ if and only if
$$\lim_{n\to \infty }P_{n}\left(a_{1},a_{2},a_{3}\right)=P\left(a_{1},a_{2},a_{3}\right),\;\forall a_{i}\in \left[o_{i}\right]\left(i=1,2,3\right).$$
Thus, the set $\mathrm{PT}(\Delta_{3})$ of all PTs over $\Omega_{3}$ forms a compact convex set in the Hilbert space $P(\Omega_{3})$.

Since the hidden variables in a C-triLHVM or a D-triLHVM for a PT are assumed to be independent, the sets ${\mathrm{PT}}^{C-\mathrm{trilocal}}\left(\Omega_{3}\right)$ and ${\mathrm{PT}}^{D-\mathrm{trilocal}}\left(\Omega_{3}\right)$ are not necessarily convex. However, we have the following.

**Proposition** **5.** 
*Both ${\mathrm{PT}}^{C-\mathrm{trilocal}}\left(\Omega_{3}\right)$ and ${\mathrm{PT}}^{D-\mathrm{trilocal}}\left(\Omega_{3}\right)$ are path-connected sets in the Hilbert space $P(\Omega_{3})$.*


**Proof.** Let $P=〚P(a_{1},a_{2},a_{3})〛$ and $Q=〚Q(a_{1},a_{2},a_{3})〛$ be any two elements of ${\mathrm{PT}}^{C-\mathrm{trilocal}}\left(\Omega_{3}\right)$. Then, P and Q have C-trLHVMs:
$$P\left(a_{1},a_{2},a_{3}\right)=\int_{\Lambda }p_{1}\left(\lambda_{1}\right)p_{2}\left(\lambda_{2}\right)p_{3}\left(\lambda_{3}\right)P_{1}\left(a_{1}|\lambda_{3}\lambda_{1}\right)P_{2}\left(a_{2}|\lambda_{1}\lambda_{2}\right)P_{3}\left(a_{3}|\lambda_{2}\lambda_{3}\right)d\mu \left(\lambda \right),$$
$$Q\left(a_{1},a_{2},a_{3}\right)=\int_{\Gamma }q_{1}\left(\xi_{1}\right)q_{2}\left(\xi_{2}\right)q_{3}\left(\xi_{3}\right)Q_{1}\left(a_{1}|\xi_{3}\xi_{1}\right)Q_{2}\left(a_{2}|\xi_{1}\xi_{2}\right)Q_{3}\left(a_{3}|\xi_{2}\xi_{3}\right)d\gamma \left(\xi \right),$$
for all possible $a_{1},a_{2},a_{3}$. Put $P_{0}\left(a_{1},a_{2},a_{3}\right)\equiv \frac{1}{o_{1}o_{2}o_{3}}$; then, $P_{0}:=〚P_{0}\left(a_{1},a_{2},a_{3}\right)〛$ is a D-trilocal (and then C-trilocal) CT over $\Omega_{3}.$ For every t∈[0,1/2], set
$$P_{1}^{t}\left(a_{1}|\lambda_{3}\lambda_{1}\right)=\left(1-2t\right)P_{1}\left(a_{1}|\lambda_{3}\lambda_{1}\right)+2t\frac{1}{o_{1}};$$
$$P_{2}^{t}\left(a_{2}|\lambda_{1}\lambda_{2}\right)=\left(1-2t\right)P_{2}\left(a_{2}|\lambda_{1}\lambda_{2}\right)+2t\frac{1}{o_{2}};$$
$$P_{3}^{t}\left(a_{3}|\lambda_{2}\lambda_{3}\right)=\left(1-2t\right)P_{3}\left(a_{3}|\lambda_{2}\lambda_{3}\right)+2t\frac{1}{o_{3}},$$
which are clearly PDs of $a_{1},a_{2}$ and $a_{3}$, respectively. Putting
$$P^{t}\left(a_{1},a_{2},a_{3}\right)=\int_{\Lambda }q_{1}\left(\lambda_{1}\right)q_{2}\left(\lambda_{2}\right)q_{3}\left(\lambda_{3}\right)P_{1}^{t}\left(a_{1}|\lambda_{3}\lambda_{1}\right)P_{2}^{t}\left(a_{2}|\lambda_{1}\lambda_{2}\right)P_{3}^{t}\left(a_{3}|\lambda_{2}\lambda_{3}\right)d\mu \left(\lambda \right),$$
then $P\left(t\right):=〚P^{t}\left(a_{1},a_{2},a_{3}\right)〛$ is a C-trilocal CT for all t∈[0,1/2] with P(0)=P and $P\left(1/2\right)=P_{0}$. Obviously, the map t↦P(t) from [0,1/2] into ${\mathrm{PT}}^{C-\mathrm{trilocal}}\left(\Omega_{3}\right)$ is continuous.For every t∈[1/2,1], set
$$Q_{1}^{t}\left(a_{1}|\xi_{3}\xi_{1}\right)=\left(2t-1\right)Q_{1}\left(a_{1}|\xi_{3}\xi_{1}\right)+2\left(1-t\right)\frac{1}{o_{1}};$$
$$Q_{2}^{t}\left(a_{2}|\xi_{1}\xi_{2}\right)=\left(2t-1\right)Q_{2}\left(a_{2}|\xi_{1}\xi_{2}\right)+2\left(1-t\right)\frac{1}{o_{2}};$$
$$Q_{3}^{t}\left(a_{3}|\xi_{2}\xi_{3}\right)=\left(2t-1\right)Q_{3}\left(a_{3}|\xi_{2}\xi_{3}\right)+2\left(1-t\right)\frac{1}{o_{3}},$$
which are clearly PDs of $a_{1},a_{2}$ and $a_{3}$, respectively. Putting
$$Q^{t}\left(a_{1},a_{2},a_{3}\right)=\int_{\Gamma }q_{1}\left(\xi_{1}\right)q_{2}\left(\xi_{2}\right)q_{3}\left(\xi_{3}\right)Q_{1}^{t}\left(a_{1}|\xi_{3}\xi_{1}\right)Q_{2}^{t}\left(a_{2}|\xi_{1}\xi_{2}\right)Q_{3}^{t}\left(a_{3}|\xi_{2}\xi_{3}\right)d\gamma \left(\xi \right),$$
then $Q\left(t\right):=〚Q^{t}\left(a_{1},a_{2},a_{3}\right)〛$ is a C-trilocal CT for all t∈[1/2,1] with $Q\left(1/2\right)=P_{0}$ and Q(1)=Q. Obviously, the map t↦Q(t) from [1/2,1] into ${\mathrm{PT}}^{C-\mathrm{trilocal}}\left(\Omega_{3}\right)$ is continuous.Next, we define a mapping $f:\left[0,1\right]\to {\mathrm{PT}}^{C-\mathrm{trilocal}}\left(\Omega_{3}\right)$ by
$$f\left(t\right)=\left\{\begin{array}{cc} P(t), & t\in [0,1/2];\\ Q(t), & t\in (1/2,1]. \end{array}\right.$$
Clearly, *f* is continuous everywhere and and then induces a path in ${\mathrm{PT}}^{C-\mathrm{trilocal}}\left(\Omega_{3}\right)$, connecting P and Q. This shows that ${\mathrm{PT}}^{C-\mathrm{trilocal}}\left(\Omega_{3}\right)$ is path-connected. Similarly, ${\mathrm{PT}}^{D-\mathrm{trilocal}}\left(\Omega_{3}\right)$ is also path-connected. The proof is completed. □

Clearly, if a PT is D-trilocal, then it must be C-trilocal with a C-triLHVM given by counting measures on $\Lambda_{j}\left(j=1,2,3\right)$. We can not show that the converse of this implication, but we obtain the following approximation result.

**Proposition** **6.** 
*Suppose that $P=〚P(a_{1},a_{2},a_{3})〛$ is a C-trilocal PT over $\Omega_{3}$ with a C-triLHVM given by three-hold Riemann integral over $\Lambda =\left[r_{1},s_{1}\right]\times \left[r_{2},s_{2}\right]\times \left[r_{3},s_{3}\right]$; then, P is in the closure of ${\mathrm{PT}}^{D-\mathrm{trilocal}}\left(\Omega_{3}\right)$ in the Hilbert space $P(\Omega_{3})$.*


**Proof.** Suppose that
(15)$$\begin{array}{ccc} P(a_{1},a_{2},a_{3}) & = & {\;\int \int \int }_{\Lambda }q_{1}\left(\lambda_{1}\right)q_{2}\left(\lambda_{2}\right)q_{3}\left(\lambda_{3}\right)P_{1}\left(a_{1}|\lambda_{3}\lambda_{1}\right)P_{2}\left(a_{2}|\lambda_{1}\lambda_{2}\right)\\ & & \times P_{3}\left(a_{3}|\lambda_{2}\lambda_{3}\right)d\lambda_{1}d\lambda_{2}d\lambda_{3} \end{array}$$
for all $a_{k}\in \left[o_{k}\right]\left(k=1,2,3\right)$, where $q_{k}\left(\lambda_{k}\right)\geq 0\left(\forall \lambda_{k}\in \Lambda_{k}:=\left[r_{k},s_{k}\right]\right)$ with $\int_{r_{k}}^{s_{k}}q_{k}\left(\lambda_{k}\right)d\lambda_{k}=1\left(k=1,2,3\right).$ Let us show that there exists a sequence ${\left\{P_{n}\right\}}_{n=1}^{+\infty }$ of D-trilocal PTs over $\Omega_{3}$ such that $P_{n}\to P$ as n→∞.Dividing each interval $\left[r_{k},s_{k}\right]$ into *n* small equal-length intervals:
$$I_{j}^{\left(k\right)}:=\left[r_{k}+\left(s_{k}-r_{k}\right)\left(j-1\right)/n,r_{k}+\left(s_{k}-r_{k}\right)j/n\right]\left(j=1,2,\ldots ,n\right),$$
we obtain a partition $T^{n}$ of Λ:
$$T^{n}=\left\{T_{j_{1},j_{2},j_{3}}^{n}:=I_{j_{1}}^{\left(1\right)}\times I_{j_{2}}^{\left(2\right)}\times I_{j_{3}}^{\left(3\right)}|1\leq j_{k}\leq n\left(k=1,2,3\right)\right\}.$$
For each $\left(j_{1},j_{2},j_{3}\right)\in {\left[n\right]}^{3}$, by taking a point $c_{j_{1},j_{2},j_{3}}^{n}=\left(\xi_{j_{1}}^{\left(n\right)},\xi_{j_{2}}^{\left(n\right)},\xi_{j_{3}}^{\left(n\right)}\right)\in T_{j_{1},j_{2},j_{3}}^{n}$ and letting
$$f_{n,k}=\sum_{i_{k}\in \left[n\right]}q_{k}\left(\xi_{i_{k}}^{\left(n\right)}\right),\pi_{k}^{\left(n\right)}\left(j_{k}\right)=\left\{\begin{array}{cc} \frac{q_{k}\left(\xi_{j_{k}}^{\left(n\right)}\right)}{f_{n,k}}, & \;\mathrm{if}\;f_{n,k}>0;\\ \frac{1}{n}, & \;\mathrm{if}\;f_{n,k}=0, \end{array}\right.$$
we obtain a PD ${\left\{\pi_{k}^{\left(n\right)}\left(j_{k}\right)\right\}}_{j_{k}\in \left[n\right]}$ such that
(16)$$q_{k}\left(\xi_{j_{k}}^{\left(n\right)}\right)=f_{n,k}\pi_{k}^{\left(n\right)}\left(j_{k}\right)=\pi_{k}^{\left(n\right)}\left(j_{k}\right)\sum_{i_{k}\in \left[n\right]}q_{k}\left(\xi_{i_{k}}^{\left(n\right)}\right).$$
Put
$$P_{1}^{\left(n\right)}\left(a_{1}|j_{3}j_{1}\right)=P_{1}\left(a_{1}|\xi_{j_{3}}^{\left(n\right)}\xi_{j_{1}}^{\left(n\right)}\right),P_{2}^{\left(n\right)}\left(a_{2}|j_{1}j_{2}\right)=P_{2}\left(a_{2}|\xi_{j_{1}}^{\left(n\right)}\xi_{j_{2}}^{\left(n\right)}\right),P_{3}^{\left(n\right)}\left(a_{3}|j_{2}j_{3}\right)=P_{3}\left(a_{3}|\xi_{j_{2}}^{\left(n\right)}\xi_{j_{3}}^{\left(n\right)}\right),$$
$$P_{n}\left(a_{1},a_{2},a_{3}\right)=\sum_{j_{1},j_{2},j_{3}=1}^{n}\pi_{1}^{\left(n\right)}\left(j_{1}\right)\pi_{2}^{\left(n\right)}\left(j_{2}\right)\pi_{3}^{\left(n\right)}\left(j_{3}\right)P_{1}^{\left(n\right)}\left(a_{1}|j_{3}j_{1}\right)P_{2}^{\left(n\right)}\left(a_{2}|j_{1}j_{2}\right)P_{3}^{\left(n\right)}\left(a_{3}|j_{2}j_{3}\right).$$
Clearly, $P_{n}:=〚P_{n}\left(a_{1},a_{2},a_{3}\right)〛\left(n=1,2,\ldots \right)$ are D-trilocal PTs over $\Omega_{3}$. We see from the property of Riemann integral that
(17)$$\lim_{n\to +\infty }\frac{s_{k}-r_{k}}{n}\sum_{i_{k}\in \left[n\right]}q_{k}\left(\xi_{i_{k}}^{\left(n\right)}\right)=\int_{r_{k}}^{s_{k}}q_{k}\left(\lambda_{k}\right)d\lambda_{k}=1\left(k=1,2,3\right).$$
Thus, by using Equations (Equation 17), (Equation 16) and the property of Riemann integral as well as Equation (Equation 15), we obtain that, for each $a_{k}\in \left[o_{k}\right]\left(k=1,2,3\right)$,
$$\begin{array}{ccc} & & \lim_{n\to +\infty }P_{n}\left(a_{1},a_{2},a_{3}\right)\\ & = & \lim_{n\to +\infty }\sum_{j_{1},j_{2},j_{3}=1}^{n}\pi_{1}^{\left(n\right)}\left(j_{1}\right)\pi_{2}^{\left(n\right)}\left(j_{2}\right)\pi_{3}^{\left(n\right)}\left(j_{3}\right)P_{1}^{\left(n\right)}\left(a_{1}|j_{3}j_{1}\right)P_{2}^{\left(n\right)}\left(a_{2}|j_{1}j_{2}\right)P_{3}^{\left(n\right)}\left(a_{3}|j_{2}j_{3}\right)\\ & = & \lim_{n\to +\infty }\frac{\left(s_{1}-r_{1}\right)\left(s_{2}-r_{2}\right)\left(s_{3}-r_{3}\right)}{n^{3}}\sum_{i_{1}\in \left[n\right]}q_{1}\left(\xi_{i_{1}}^{\left(n\right)}\right)\sum_{i_{2}\in \left[n\right]}q_{2}\left(\xi_{i_{2}}^{\left(n\right)}\right)\sum_{i_{3}\in \left[n\right]}q_{3}\left(\xi_{i_{3}}^{\left(n\right)}\right)\\ & & \times \sum_{j_{1},j_{2},j_{3}=1}^{n}\pi_{1}^{\left(n\right)}\left(j_{1}\right)\pi_{2}^{\left(n\right)}\left(j_{2}\right)\pi_{3}^{\left(n\right)}\left(j_{3}\right)P_{1}^{\left(n\right)}\left(a_{1}|j_{3}j_{1}\right)P_{2}^{\left(n\right)}\left(a_{2}|j_{1}j_{2}\right)P_{3}^{\left(n\right)}\left(a_{3}|j_{2}j_{3}\right)\\ & = & \lim_{n\to +\infty }\frac{\left(s_{1}-r_{1}\right)\left(s_{2}-r_{2}\right)\left(s_{3}-r_{3}\right)}{n^{3}}\sum_{j_{1},j_{2},j_{3}=1}^{n}q_{1}\left(\xi_{j_{1}}^{\left(n\right)}\right)q_{2}\left(\xi_{j_{2}}^{\left(n\right)}\right)q_{3}\left(\xi_{j_{3}}^{\left(n\right)}\right)\\ & & \times P_{1}\left(a_{1}|\xi_{j_{3}}^{\left(n\right)}\xi_{j_{1}}^{\left(n\right)}\right)P_{2}\left(a_{2}|\xi_{j_{1}}^{\left(n\right)}\xi_{j_{2}}^{\left(n\right)}\right)P_{3}\left(a_{3}|\xi_{j_{2}}^{\left(n\right)}\xi_{j_{3}}^{\left(n\right)}\right)\\ & = & {\;\int \int \int }_{\Lambda }q_{1}\left(\lambda_{1}\right)q_{2}\left(\lambda_{2}\right)q_{3}\left(\lambda_{3}\right)P_{1}\left(a_{1}|\lambda_{3}\lambda_{1}\right)P_{2}\left(a_{2}|\lambda_{1}\lambda_{2}\right)P_{3}\left(a_{3}|\lambda_{2}\lambda_{3}\right)d\lambda_{1}d\lambda_{2}d\lambda_{3}\\ & = & P(a_{1},a_{2},a_{3}). \end{array}$$
This shows that $P_{n}\to P$ as n→∞. The proof is completed. □

This conclusion implies that, if the set of all a *D*-trilocal PTs $P=〚P(a_{1},a_{2},a_{3})〛$ over $\Omega_{3}$ is closed, then the PT given by Equation (Equation 15) is *D*-trilocal.

In addition, when a PT P is given by Equation (Equation 15) where $\Lambda =\left[s_{1},+\infty \right)\times \left[s_{2},+\infty \right)\times \left[s_{3},+\infty \right)$, DFs $q_{i}$ and RFs $P_{i}\left(a_{i}|\cdot \cdot \right)$ are Riemann integrable on any $\left[s_{i},S_{i}\right]$ and $\left[s_{1},S_{1}\right]\times \left[s_{2},S_{2}\right]\times \left[s_{3},S_{3}\right]$, respectively, it is C-trilocal with a C-triLHVM (Equation 15) given by Lebesgue measure on Λ. In this case, the Levi’s lemma yields that
(18)$$\begin{array}{ccc} P(a_{1},a_{2},a_{3}) & = & \lim_{n\to +\infty }{\;\int \int \int }_{\Lambda_{n}}q_{1}\left(\lambda_{1}\right)q_{2}\left(\lambda_{2}\right)q_{3}\left(\lambda_{3}\right)P_{1}\left(a_{1}|\lambda_{3}\lambda_{1}\right)\\ & & \times P_{2}\left(a_{2}|\lambda_{1}\lambda_{2}\right)P_{3}\left(a_{3}|\lambda_{2}\lambda_{3}\right)d\lambda_{1}d\lambda_{2}d\lambda_{3} \end{array}$$
for all $a_{k}\in \left[o_{k}\right]\left(k=1,2,3\right)$, where $\Lambda_{n}=\left[s_{1},s_{1}+n\right]\times \left[s_{2},s_{2}+n\right]\times \left[s_{3},s_{3}+n\right]$. Put
$$q_{i}^{\left(n\right)}\left(\lambda_{i}\right)=\frac{q_{i}\left(\lambda_{i}\right)}{\int_{\left[s_{i},s_{i}+n\right]}q_{i}\left(t_{i}\right)dt_{i}}\left(n=1,2,\ldots \right),$$
then $\lim_{n\to +\infty }\int_{\left[s_{i},s_{i}+n\right]}q_{i}\left(t_{i}\right)dt_{i}=\int_{\left[s_{i},+\infty \right)}q_{i}\left(t_{i}\right)dt_{i}=1$ as n→+∞, and
$$q_{i}^{\left(n\right)}\left(\lambda_{i}\right)\geq 0,\forall \lambda_{i}\in \left[s_{i},s_{i}+n\right],\int_{\left[s_{i},s_{i}+n\right]}q_{i}^{\left(n\right)}\left(\lambda_{i}\right)d\lambda_{i}=1.$$
For each n=1,2,…, letting
(19)$$\begin{array}{ccc} P_{n}\left(a_{1},a_{2},a_{3}\right) & = & {\;\int \int \int }_{\Lambda_{n}}q_{1}^{\left(n\right)}\left(\lambda_{1}\right)q_{2}^{\left(n\right)}\left(\lambda_{2}\right)q_{3}^{\left(n\right)}\left(\lambda_{3}\right)P_{1}\left(a_{1}|\lambda_{3}\lambda_{1}\right)\\ & & \times P_{2}\left(a_{2}|\lambda_{1}\lambda_{2}\right)P_{3}\left(a_{3}|\lambda_{2}\lambda_{3}\right)d\lambda_{1}d\lambda_{2}d\lambda_{3}, \end{array}$$
we obtain a C-trilocal PT $P_{n}=〚P_{n}\left(a_{1},a_{2},a_{3}\right)〛$ over $\Omega_{3}$ with a C-triLHVM (Equation 19) in terms of Riemann integral over $\Lambda_{n}$. Proposition 6 yields that $P_{n}\in \bar{{\mathrm{PT}}^{D-\mathrm{trilocal}}\left(\Omega_{3}\right)}$ for all *n*. Equation (Equation 18) implies that $P=\lim_{n\to +\infty }P_{n}$. It follows that $P\in \bar{{\mathrm{PT}}^{D-\mathrm{trilocal}}\left(\Omega_{3}\right)}$.

Similarly, one can check that the PT P over $\Omega_{3}$ defined by infinite series
$$P\left(a_{1},a_{2},a_{3}\right)=\sum_{\lambda_{1}=s_{1}}^{+\infty }\sum_{\lambda_{2}=s_{2}}^{+\infty }\sum_{\lambda_{3}=s_{3}}^{+\infty }q_{1}\left(\lambda_{1}\right)q_{2}\left(\lambda_{2}\right)q_{3}\left(\lambda_{3}\right)P_{1}\left(a_{1}|\lambda_{3}\lambda_{1}\right)P_{2}\left(a_{2}|\lambda_{1}\lambda_{2}\right)P_{3}\left(a_{3}|\lambda_{2}\lambda_{3}\right)$$
is also C-trilocal and in the closure $\bar{{\mathrm{PT}}^{D-\mathrm{trilocal}}\left(\Omega_{3}\right)}$ of ${\mathrm{PT}}^{D-\mathrm{trilocal}}\left(\Omega_{3}\right)$.

## 3. Trilocality of Tripartite CTs

In this section, we aim to discuss two types of trilocality of a tripartite correlation tensor (CTs) [35]: $P=〚P(a_{1}a_{2}a_{3}|x_{1}x_{2}x_{3})〛$ over an index set
$$\Delta_{3}=\left[o_{1}\right]\times \left[o_{2}\right]\times \left[o_{3}\right]\times \left[m_{1}\right]\times \left[m_{2}\right]\times \left[m_{3}\right],$$
which is a nonnegative tensor with index set $\Delta_{3}$ such that
$$\sum_{a_{i}\in \left[o_{i}\right]}P\left(a_{1}a_{2}a_{3}|x_{1}x_{2}x_{3}\right)=1,\;\;\forall x_{i}\in \left[m_{i}\right]\left(i=1,2,3\right).$$
We use $\mathrm{CT}(\Delta_{3})$ to denote the sets of CTs over $\Delta_{3}$.

**Definition** **4.** 
*Let $P=〚P(a_{1}a_{2}a_{3}|x_{1}x_{2}x_{3})〛$ be a CT over $\Delta_{3}.$*


(1) P is said to *C-trilocal* if it has a C-triLHVM:(20)$$\begin{array}{ccc} P(a_{1}a_{2}a_{3}|x_{1}x_{2}x_{3}) & = & \int_{\Lambda }q_{1}\left(\lambda_{1}\right)q_{2}\left(\lambda_{2}\right)q_{3}\left(\lambda_{3}\right)P_{1}\left(a_{1}|x_{1},\lambda_{3}\lambda_{1}\right)\\ & & \times P_{2}\left(a_{2}|x_{2},\lambda_{1}\lambda_{2}\right)P_{3}\left(a_{3}|x_{3},\lambda_{2}\lambda_{3}\right)d\mu \left(\lambda \right) \end{array}$$
for a product measure space
$$\left(\Lambda ,\Omega ,\mu \right)=\left(\Lambda_{1}\times \Lambda_{2}\times \Lambda_{3},\Omega_{1}\times \Omega_{2}\times \Omega_{3},\mu_{1}\times \mu_{2}\times \mu_{3}\right),$$
where $\lambda =(\lambda_{1},\lambda_{2},\lambda_{3})$, $q_{j}\left(\lambda_{j}\right)$ is a DF of $\lambda_{j}$, $P_{1}\left(a_{1}|x_{1},\lambda_{3}\lambda_{1}\right),P_{2}\left(a_{2}|x_{2},\lambda_{1}\lambda_{2}\right)$ and $P_{3}\left(a_{3}|x_{3},\lambda_{2}\lambda_{3}\right)$, called response functions (RSs) at nodes 1,2 and 3, are nonnegative Ω-measurable on Λ for all $x_{i},a_{i}$ and PDs of outcomes $a_{1},a_{2}$ and $a_{3}$, respectively, for all $\lambda_{1},\lambda_{2}$ and $\lambda_{3}$.

(2) P is said to be *D-trilocal* if it has a D-triLHVM:(21)$$\begin{array}{ccc} P(a_{1}a_{2}a_{3}|x_{1}x_{2}x_{3}) & = & \sum_{\lambda_{1}=1}^{n_{1}}\sum_{\lambda_{2}=1}^{n_{2}}\sum_{\lambda_{3}=1}^{n_{3}}q_{1}\left(\lambda_{1}\right)q_{2}\left(\lambda_{2}\right)q_{3}\left(\lambda_{3}\right)P_{1}\left(a_{1}|x_{1},\lambda_{3}\lambda_{1}\right)\\ & & \times P_{2}\left(a_{2}|x_{2},\lambda_{1}\lambda_{2}\right)P_{3}\left(a_{3}|x_{3},\lambda_{2}\lambda_{3}\right) \end{array}$$
for all $x_{k}\in \left[m_{k}\right],a_{k}\in \left[o_{k}\right]\left(k=1,2,3\right)$, where
$$q_{k}\left(\lambda_{k}\right),P_{1}\left(a_{1}|x_{1},\lambda_{3}\lambda_{1}\right),P_{2}\left(a_{2}|x_{2},\lambda_{1}\lambda_{2}\right),P_{3}\left(a_{3}|x_{3},\lambda_{2}\lambda_{3}\right)$$
are PDs of $\lambda_{k},a_{1},a_{2},a_{3}$, respectively.

(3) P is said to be *C-nontrilocal* (resp. *D -nontrilocal*) if it is not C-trilocal (resp. not D-trilocal).

We use ${\mathrm{CT}}^{C-\mathrm{trilocal}}\left(\Delta_{3}\right)$ and ${\mathrm{CT}}^{D-\mathrm{trilocal}}\left(\Delta_{3}\right)$ to denote the sets of all C-trilocal and D-trilocal CTs over $\Delta_{3}$, respectively. Clearly, ${\mathrm{CT}}^{C-\mathrm{trilocal}}\left(\Delta_{3}\right)⊃{\mathrm{CT}}^{D-\mathrm{trilocal}}\left(\Delta_{3}\right)$.

Similar to the analysis before Proposition 1, we can obtain the following.

**Proposition** **7.** 
*A CT $P=〚P(a_{1}a_{2}a_{3}|x_{1}x_{2}x_{3})〛$ over $\Delta_{3}$ is C-trilocal if and only if it admits a C-triLHVM:*

(22)
$$\begin{array}{ccc} P(a_{1}a_{2}a_{3}|x_{1}x_{2}x_{3}) & = & \int_{\Lambda }P_{1}\left(a_{1}|x_{1},\lambda_{3}\lambda_{1}\right)P_{2}\left(a_{2}|x_{2},\lambda_{1}\lambda_{2}\right)P_{3}\left(a_{3}|x_{3},\lambda_{2}\lambda_{3}\right)d\gamma \left(\lambda \right) \end{array}$$

*for some product probability space*

$$\left(\Lambda ,\Sigma ,\gamma \right)=\left(\Lambda_{1}\times \Lambda_{2}\times \Lambda_{3},\Sigma_{1}\times \Sigma_{2}\times \Sigma_{3},\gamma_{1}\times \gamma_{2}\times \gamma_{3}\right).$$



It is obvious that different C-trilocal CTs over the same index set $\Delta_{3}$ have their C-triLHVMs that are given by product measure spaces that may be different. However, the following result shows that a finite number of C-trilocal CTs $P_{k}\left(k=1,2,\ldots ,m\right)$ over $\Delta_{3}$ have C-triLHVMs based on a common product measure space.

**Proposition** **8.** 
*Let $P_{k}=〚P_{k}\left(a_{1}a_{2}a_{3}|x_{1}x_{2}x_{3}\right)〛\left(k=1,2,\ldots ,m\right)$ be m C-trlocal CTs over $\Delta_{3}$. Then, there is a product measure space*

$$\left(S_{1}\times S_{2}\times S_{3},T_{1}\times T_{2}\times T_{3},\gamma_{1}\times \gamma_{2}\times \gamma_{3}\right)$$

*and three DFs $f_{i}\left(s_{i}\right)$ of $s_{i}\left(i=1,2,3\right)$ such that*

(23)
$$\begin{array}{ccc} P_{k}\left(a_{1}a_{2}a_{3}|x_{1}x_{2}x_{3}\right) & = & {\;\int \int \int }_{S_{1}\times S_{2}\times S_{3}}f_{1}\left(s_{1}\right)f_{2}\left(s_{2}\right)f_{3}\left(s_{3}\right)P_{1}^{\left(k\right)}\left(a_{1}|x_{1},s_{3}s_{1}\right)P_{2}^{\left(k\right)}\left(a_{2}|x_{2},s_{1}s_{2}\right)\\ & & \times P_{3}^{\left(k\right)}\left(a_{3}|x_{3},s_{2}s_{3}\right)d\gamma_{1}\left(s_{1}\right)d\gamma_{2}\left(s_{2}\right)d\gamma_{3}\left(s_{3}\right),\;\forall k\in \left[m\right], \end{array}$$

*for all $a_{i},x_{i}.$*


**Proof.** By Definition 4, each $P_{k}$ can be represented as
(24)$$\begin{array}{ccc} P_{k}\left(a_{1}a_{2}a_{3}|x_{1}x_{2}x_{3}\right) & = & {\;\int \int \int }_{\Lambda_{1}^{\left(k\right)}\times \Lambda_{2}^{\left(k\right)}\times \Lambda_{3}^{\left(k\right)}}q_{1}^{\left(k\right)}\left(\lambda_{1}^{\left(k\right)}\right)q_{2}^{\left(k\right)}\left(\lambda_{2}^{\left(k\right)}\right)q_{3}^{\left(k\right)}\left(\lambda_{3}^{\left(k\right)}\right)P_{A}^{\left(k\right)}\left(a_{1}|x_{1},\lambda_{3}^{\left(k\right)}\lambda_{1}^{\left(k\right)}\right)\\ & & \times P_{B}^{\left(k\right)}\left(a_{2}|x_{2},\lambda_{1}^{\left(k\right)}\lambda_{2}^{\left(k\right)}\right)P_{C}^{\left(k\right)}\left(a_{3}|x_{3},\lambda_{2}^{\left(k\right)}\lambda_{3}^{\left(k\right)}\right)\\ & & \times d\mu_{1}^{\left(k\right)}\left(\lambda_{1}^{\left(k\right)}\right)d\mu_{2}^{\left(k\right)}\left(\lambda_{2}^{\left(k\right)}\right)d\mu_{3}^{\left(k\right)}\left(\lambda_{3}^{\left(k\right)}\right) \end{array}$$
for some product measure space
$$(\Lambda_{1}^{\left(k\right)}\times \Lambda_{2}^{\left(k\right)}\times \Lambda_{3}^{\left(k\right)},\Omega_{1}^{\left(k\right)}\times \Omega_{2}^{\left(k\right)}\times \Omega_{3}^{\left(k\right)},\mu_{1}^{\left(k\right)}\times \mu_{2}^{\left(k\right)}\times \mu_{3}^{\left(k\right)}).$$
Putting
$$S_{i}=\prod_{k=1}^{m}\Lambda_{i}^{\left(k\right)},T_{i}=\prod_{k=1}^{m}\Omega_{i}^{\left(k\right)},\gamma_{i}=\prod_{k=1}^{m}\mu_{i}^{\left(k\right)},$$
$$s_{i}=\left(\lambda_{i}^{\left(1\right)},\lambda_{i}^{\left(2\right)},\ldots ,\lambda_{i}^{\left(m\right)}\right),f_{i}\left(s_{i}\right)=\prod_{k=1}^{m}q_{i}^{\left(k\right)}\left(\lambda_{i}^{\left(k\right)}\right)\left(i=1,2,3\right)$$
produces a product measure space
$$\left(S_{1}\times S_{2}\times S_{3},T_{1}\times T_{2}\times T_{3},\gamma_{1}\times \gamma_{2}\times \gamma_{3}\right)$$
and three DFs $f_{i}\left(s_{i}\right)$ of $s_{i}\left(i=1,2,3\right)$. By letting
$$P_{1}^{\left(k\right)}\left(a_{1}|x_{1},s_{3}s_{1}\right)=P_{A}^{\left(k\right)}\left(a_{1}|x_{1},\lambda_{3}^{\left(k\right)}\lambda_{1}^{\left(k\right)}\right),$$
$$P_{2}^{\left(k\right)}\left(a_{2}|x_{2},s_{1}s_{2}\right))=P_{B}^{\left(k\right)}\left(a_{2}|x_{2},\lambda_{1}^{\left(k\right)}\lambda_{2}^{\left(k\right)}\right),$$
$$P_{3}^{\left(k\right)}\left(a_{3}|x_{3},s_{2}s_{3}\right)=P_{C}^{\left(k\right)}\left(a_{3}|x_{3},\lambda_{2}^{\left(k\right)}\lambda_{3}^{\left(k\right)}\right),$$
for all $s_{i}=\left(\lambda_{i}^{\left(1\right)},\lambda_{i}^{\left(2\right)},\ldots ,\lambda_{i}^{\left(m\right)}\right)$ in $S_{i}$, we obtain (Equation 23) using Equation (Equation 24). The proof is completed.Using Definitions 1 and 4, we see that, when a CT $P=〚P(a_{1}a_{2}a_{3}|x_{1}x_{2}x_{3})〛$ over $\Delta_{3}$ is C-trilocal (resp. D-trilocal), the induced PTs $P_{x_{1}x_{2}x_{3}}:=〚P\left(a_{1}a_{2}a_{3}|x_{1}x_{2}x_{3}\right)〛$ over $\Omega_{3}$ must be C-trilocal (resp. D-trilocal) for all $\left(x_{1},x_{2},x_{3}\right)$ in $\left[m_{1}\right]\times \left[m_{2}\right]\times \left[m_{3}\right]$. Equivalently, if the PT $P_{x_{1}^{0}x_{2}^{0}x_{3}^{0}}$ is non-C-trilocal (resp. non-D-trilocal) for some $\left(x_{1}^{0},x_{2}^{0},x_{3}^{0}\right)$ in $\left[m_{1}\right]\times \left[m_{2}\right]\times \left[m_{3}\right]$, then the CT $P=〚P(a_{1}a_{2}a_{3}|x_{1}x_{2}x_{3})〛$ must be non-C-trilocal (resp. non-D-trilocal). In this sense, we can say that the non-trilocality of PTs is stronger than that of CTs. Furthermore, let $P=〚P(a_{1}a_{2}a_{3}|x_{1}x_{2}x_{3})〛$ be a C-trilocal CT. Then, it has a C-triLHVM (Equation 20). By letting
$$\begin{array}{ccc} P_{1}\left(a_{1}|x_{1},\lambda_{1}\right) & = & \int_{\Lambda_{3}}q_{3}\left(\lambda_{3}\right)P_{1}\left(a_{1}|x_{1},\lambda_{3}\lambda_{1}\right)d\mu_{3}\left(\lambda_{3}\right);\\ P_{2}\left(a_{2}|x_{2},\lambda_{1}\right) & = & \int_{\Lambda_{2}}q_{2}\left(\lambda_{2}\right)P_{2}\left(a_{2}|x_{2},\lambda_{1}\lambda_{2}\right)d\mu_{2}\left(\lambda_{2}\right), \end{array}$$
we see from (Equation 20) that the marginal distribution of P on the subsystem $S_{1}S_{2}$ reads
(25)$$\begin{array}{c} P_{12}\left(a_{1}a_{2}|x_{1}x_{2}\right)=\sum_{a_{3}}P\left(a_{1}a_{2}a_{3}|x_{1}x_{2}x_{3}\right)=\int_{\Lambda_{1}}q_{1}\left(\lambda_{1}\right)P_{1}\left(a_{1}|x_{1},\lambda_{1}\right)P_{2}\left(a_{2}|x_{2},\lambda_{1}\right)d\mu_{1}\left(\lambda_{1}\right) \end{array}$$
for all possible $x_{1},x_{2},a_{1},a_{2}$. Thus, $P_{12}=〚P_{12}\left(a_{1}a_{2}|x_{1}x_{2}\right)〛$ becomes a Bell local CT [35] over $\left[o_{1}\right]\times \left[o_{2}\right]\times \left[m_{1}\right]\times \left[m_{2}\right].$ Similarly, the marginal distributions $P_{23}=〚P_{23}\left(a_{2}a_{3}|x_{2}x_{3}\right)〛$ and $P_{13}=〚P_{13}\left(a_{1}a_{3}|x_{1}x_{3}\right)〛$ are Bell local CTs over $\left[o_{2}\right]\times \left[o_{3}\right]\times \left[m_{2}\right]\times \left[m_{3}\right]$ and $\left[o_{1}\right]\times \left[o_{3}\right]\times \left[m_{1}\right]\times \left[m_{3}\right]$, respectively. This analysis leads to the following necessary condition for a CT to be C-trilocal. □

**Proposition** **9.** 
*The three bipartite marginal distributions of a tripartite C-trilocal CT are Bell local.*


**Remark** **1.** 
*In particular, when $\Lambda_{3}$ is a singleton $\left\{\lambda_{3}\right\}\left(\lambda_{3}=1\right)$ and $q_{3}\left(\lambda_{3}\right)=\mu_{3}\left(\left\{\lambda_{3}\right\}\right)=1$, Equation (Equation 20) becomes*

(26)
$$\begin{array}{ccc} P(a_{1}a_{2}a_{3}|x_{1}x_{2}x_{3}) & = & {\;\int \int }_{\Lambda_{1}\times \Lambda_{2}}q_{1}\left(\lambda_{1}\right)q_{2}\left(\lambda_{2}\right)P_{1}\left(a_{1}|x_{1},\lambda_{1}\right)P_{2}\left(a_{2}|x_{2},\lambda_{1}\lambda_{2}\right)\\ & & \times P_{3}\left(a_{3}|x_{3},\lambda_{2}\right)d\mu_{1}\left(\lambda_{1}\right)d\mu_{2}\left(\lambda_{2}\right). \end{array}$$

*In this case, P is said to be C-bilocal, shortly bilocal [20,21,36] and Equation (Equation 26) is called a C-biLHVM of P. In addition, when $\Lambda_{2}$ and $\Lambda_{3}$ can be chosen as finite sets, P is said to be D-bilocal. We use ${\mathrm{CT}}^{C-\mathrm{bilocal}}\left(\Delta_{3}\right)$ and ${\mathrm{CT}}^{D-\mathrm{bilocal}}\left(\Delta_{3}\right)$ to denote the sets of all C-bilocal and D-bilocal CTs over $\Delta_{3}$, respectively. Conversely, when P is a C-bilocal over $\Delta_{3}$, it has a C-biLHVM (Equation 26), which can be written as (Equation 20) with $\Lambda_{3}$ being a singleton $\left\{\lambda_{3}\right\}$ with $\lambda_{3}=1$ and $q_{3}\left(\lambda_{3}\right)=\mu_{3}\left(\left\{\lambda_{3}\right\}\right)=1$. Thus,*

$${\mathrm{CT}}^{C-\mathrm{bilocal}}\left(\Delta_{3}\right)\subset {\mathrm{CT}}^{C-\mathrm{trilocal}}\left(\Delta_{3}\right),\;{\mathrm{CT}}^{D-\mathrm{bilocal}}\left(\Delta_{3}\right)\subset {\mathrm{CT}}^{D-\mathrm{trilocal}}\left(\Delta_{3}\right).$$

*It is proved in ([36] Theorem 2.1) that*

$${\mathrm{CT}}^{C-\mathrm{bilocal}}\left(\Delta_{3}\right)={\mathrm{CT}}^{D-\mathrm{bilocal}}\left(\Delta_{3}\right):={\mathrm{CT}}^{\mathrm{bilocal}}\left(\Delta_{3}\right).$$



**Definition** **5.** 
*A tripartite CT $P=〚P(a_{1}a_{2}a_{3}|x_{1}x_{2}x_{3})〛$ over $\Delta_{3}$ is said to be tri-quantum if there exists a TN with the state $\rho_{\mathrm{TN}}$ and a set of local POVMs*

(27)
$$M=\left\{M_{x_{1}x_{2}x_{3}}|x_{k}\in \left[m_{k}\right]\right\}=\left\{M_{x_{1}}^{\left(1\right)}⊗M_{x_{2}}^{\left(2\right)}⊗M_{x_{3}}^{\left(3\right)}|x_{k}\in \left[m_{k}\right]\right\},$$

*with $M_{x_{k}}^{\left(k\right)}={\left\{M_{a_{k}|x_{k}}^{\left(k\right)}\right\}}_{a_{k}=1}^{o_{k}}$ such that $P=T_{TN}^{M}$, where*

(28)
$$\begin{array}{c} T_{TN}^{M}\left(a_{1}a_{2}a_{3}|x_{1}x_{2}x_{3}\right)=\mathrm{tr}\left[\left({⊗}_{n=1}^{3}M_{a_{n}|x_{k}}^{\left(n\right)}\right)\tilde{\rho_{\mathrm{TN}}}\right],\;\forall a_{k}\in \left[o_{k}\right] \end{array}$$

*for all possible $x_{k},a_{k}$. In particular, when the shares states $\rho_{i,j}$ can be chosen as separable states, we say that P is separable tri-quantum.*


**Definition** **6.**
*A triangle network TN given by Figure 1 is said to be strongly trilocal if, for any set M of local POVMs of the form (Equation 27), the resulting CT $T_{TN}^{M}$ is D-trilocal.*


Using Proposition 9, we see that, when one of the three marginal distributions is Bell nonlocal, P must be neither C-trilocal nor D-trilocal. Since every entangled pure state is Bell nonlocal [37], when one of the shared states $\rho_{i,j}$ in the triangle network given by Figure 1 is an entangled pure state, there are a set of local POVMs (Equation 27) such that the resulting CT $P=T_{TN}^{M}$ is not C-trilocal and then not D-trilocal. Thus, the network is not strongly trilocal. Conversely, we have the following.

**Proposition** **10.**
*Every separable (i.e., all shared states $\rho_{i,j}$ are separable) triangle network TN given by Figure 1 is strongly trilocal.*


**Proof** **.** Suppose that the TN given by Figure 1 is separable. Then, the shared states $\rho_{s,t}$ are separable, i.e., there exist PDs ${\left\{q_{1}\left(\lambda_{1}\right)\right\}}_{\lambda_{1}=1}^{n_{1}},{\left\{q_{2}\left(\lambda_{2}\right)\right\}}_{\lambda_{2}=1}^{n_{2}}$ and ${\left\{q_{3}\left(\lambda_{3}\right)\right\}}_{\lambda_{3}=1}^{n_{3}}$ such that
$$\rho_{1,2}=\sum_{\lambda_{1}=1}^{n_{1}}q_{1}\left(\lambda_{1}\right)\rho_{1}^{\left(1\right)}\left(\lambda_{1}\right)⊗\rho_{1}^{\left(2\right)}\left(\lambda_{1}\right)\in D\left(H_{1}^{\left(1\right)}⊗H_{1}^{\left(2\right)}\right),$$
$$\rho_{2,3}=\sum_{\lambda_{2}=1}^{n_{2}}q_{2}\left(\lambda_{2}\right)\rho_{2}^{\left(2\right)}\left(\lambda_{2}\right)⊗\rho_{1}^{\left(3\right)}\left(\lambda_{2}\right)\in D\left(H_{2}^{\left(2\right)}⊗H_{1}^{\left(3\right)}\right),$$
$$\rho_{3,1}=\sum_{\lambda_{3}=1}^{n_{3}}q_{3}\left(\lambda_{3}\right)\rho_{2}^{\left(3\right)}\left(\lambda_{3}\right)⊗\rho_{2}^{\left(1\right)}\left(\lambda_{3}\right)\in D\left(H_{2}^{\left(3\right)}⊗H_{2}^{\left(1\right)}\right),$$
where $\rho_{t}^{\left(s\right)}\left(r\right)\in D\left(H_{t}^{\left(s\right)}\right).$ Thus, the network state reads
$$\begin{array}{ccc} \rho_{\mathrm{TN}} & = & \sum_{\lambda_{1},\lambda_{2},\lambda_{3}}q_{1}\left(\lambda_{1}\right)q_{2}\left(\lambda_{2}\right)q_{3}\left(\lambda_{3}\right)\rho_{1}^{\left(1\right)}\left(\lambda_{1}\right)⊗\rho_{1}^{\left(2\right)}\left(\lambda_{1}\right)⊗\rho_{2}^{\left(2\right)}\left(\lambda_{2}\right)⊗\rho_{1}^{\left(3\right)}\left(\lambda_{2}\right)⊗\rho_{2}^{\left(3\right)}\left(\lambda_{3}\right)⊗\rho_{2}^{\left(1\right)}\left(\lambda_{3}\right), \end{array}$$
being a state of system $H_{1}^{\left(1\right)}⊗H_{1}^{\left(2\right)}⊗H_{2}^{\left(2\right)}⊗H_{1}^{\left(3\right)}⊗H_{2}^{\left(3\right)}⊗H_{2}^{\left(1\right)}$. Then, the measurement state is
$$\begin{array}{ccc} \tilde{\rho_{\mathrm{TN}}} & = & \sum_{\lambda_{1},\lambda_{2},\lambda_{3}}q_{1}\left(\lambda_{1}\right)q_{2}\left(\lambda_{2}\right)q_{3}\left(\lambda_{3}\right)\left(\rho_{2}^{\left(1\right)}\left(\lambda_{3}\right)⊗\rho_{1}^{\left(1\right)}\left(\lambda_{1}\right)\right)⊗\left(\rho_{1}^{\left(2\right)}\left(\lambda_{1}\right)⊗\rho_{2}^{\left(2\right)}\left(\lambda_{2}\right)\right)⊗\left(\rho_{1}^{\left(3\right)}\left(\lambda_{2}\right)⊗\rho_{2}^{\left(3\right)}\left(\lambda_{3}\right)\right). \end{array}$$
being a state of system
$$H^{\left(1\right)}⊗H^{\left(2\right)}⊗H^{\left(3\right)}=\left(H_{2}^{\left(1\right)}⊗H_{1}^{\left(1\right)}\right)⊗\left(H_{1}^{\left(2\right)}⊗H_{2}^{\left(2\right)}\right)⊗\left(H_{1}^{\left(3\right)}⊗H_{2}^{\left(3\right)}\right).$$
for any set M of local POVMs of the form (Equation 27) of system $H^{\left(1\right)}⊗H^{\left(2\right)}⊗H^{\left(3\right)}$, we compete that
$$\begin{array}{ccc} T_{\mathrm{TN}}^{M}\left(a_{1}a_{2}a_{3}|x_{1}x_{2}x_{3}\right) & = & \mathrm{tr}[\left({⊗}_{n=1}^{3}M_{a_{n}}^{\left(n\right)}\right)\tilde{\rho_{\mathrm{TN}}}]\\ & = & \sum_{\lambda_{1},\lambda_{2},\lambda_{3}}q_{1}\left(\lambda_{1}\right)q_{2}\left(\lambda_{2}\right)q_{3}\left(\lambda_{3}\right)\mathrm{tr}\left[M_{a_{1}|x_{1}}^{\left(1\right)}\left(\rho_{1}^{\left(1\right)}\left(\lambda_{1}\right)⊗\rho_{2}^{\left(1\right)}\left(\lambda_{3}\right)\right)\right]\\ & & \times \mathrm{tr}\left[M_{a_{2}|x_{2}}^{\left(2\right)}\left(\rho_{1}^{\left(2\right)}\left(\lambda_{1}\right)⊗\rho_{2}^{\left(2\right)}\left(\lambda_{2}\right)\right)\right]\mathrm{tr}\left[M_{a_{3}|x_{3}}^{\left(3\right)}\left(\rho_{1}^{\left(3\right)}\left(\lambda_{2}\right)⊗\rho_{2}^{\left(3\right)}\left(\lambda_{3}\right)\right)\right]\\ & = & \sum_{\lambda_{k}\in \left[n_{k}\right]}q_{1}\left(\lambda_{1}\right)q_{2}\left(\lambda_{2}\right)q_{3}\left(\lambda_{3}\right)P_{1}\left(a_{1}|x_{1},\lambda_{3}\lambda_{1}\right)P_{2}\left(a_{2}|x_{2},\lambda_{1}\lambda_{2}\right)P_{3}\left(a_{3}|x_{3},\lambda_{2}\lambda_{3}\right), \end{array}$$
for all $a_{k}\in \left[o_{k}\right]$, where
$$P_{1}\left(a_{1}|x_{1},\lambda_{3}\lambda_{1}\right)=\mathrm{tr}\left[M_{a_{1}|x_{1}}^{\left(1\right)}\left(\rho_{1}^{\left(1\right)}\left(\lambda_{1}\right)⊗\rho_{2}^{\left(1\right)}\left(\lambda_{3}\right)\right)\right],$$
$$P_{2}\left(a_{2}|x_{2},\lambda_{1}\lambda_{2}\right)=\mathrm{tr}\left[M_{a_{2}|x_{2}}^{\left(2\right)}\left(\rho_{1}^{\left(2\right)}\left(\lambda_{1}\right)⊗\rho_{2}^{\left(2\right)}\left(\lambda_{2}\right)\right)\right],$$
$$P_{3}\left(a_{3}|x_{3},\lambda_{2}\lambda_{3}\right)=\mathrm{tr}\left[M_{a_{3}|x_{3}}^{\left(3\right)}\left(\rho_{1}^{\left(3\right)}\left(\lambda_{2}\right)⊗\rho_{2}^{\left(3\right)}\left(\lambda_{3}\right)\right)\right].$$
Clearly, ${\left\{q_{k}\left(\lambda_{k}\right)\right\}}_{\lambda_{k}\in \left[n_{k}\right]},{\left\{P_{1}\left(a_{1}|x_{1},\lambda_{3}\lambda_{1}\right)\right\}}_{a_{1}\in \left[o_{1}\right]},{\left\{P_{2}\left(a_{2}|x_{2},\lambda_{1}\lambda_{2}\right)\right\}}_{a_{2}\in \left[o_{2}\right]},$ and ${\left\{P_{3}\left(a_{3}|x_{3},\lambda_{2}\lambda_{3}\right)\right\}}_{a_{3}\in \left[o_{3}\right]}$ are PDs of $\lambda_{k},a_{1},a_{2}$ and $a_{3}$, respectively. This shows that $T_{\mathrm{TN}}^{M}$ is D-trilocal. It follows from Definition 6 that the triangle network TN given by Figure 1 is strongly trilocal. The proof is completed. □

**Theorem** **1.****(Realization).***A CT*P over $\Delta_{3}$
*is D-trilocal if and only if it is separable tri-quantum.*

**Proof.** The sufficiency is given by Proposition 10. To show the necessity, we let $P=\{P\left(a_{1}a_{2}a_{3}|x_{1}x_{2}x_{3}\right)\}$ be a D-trilocal PT over $\Delta_{3}$. Then, it can be written as the form of (Equation 21):
(29)$$\begin{array}{ccc} P(a_{1}a_{2}a_{3}|x_{1}x_{2}x_{3}) & = & \sum_{\lambda_{1}=1}^{n_{1}}\sum_{\lambda_{2}=1}^{n_{2}}\sum_{\lambda_{3}=1}^{n_{3}}q_{1}\left(\lambda_{1}\right)q_{2}\left(\lambda_{2}\right)q_{3}\left(\lambda_{3}\right)P_{1}\left(a_{1}|x_{1},\lambda_{3}\lambda_{1}\right)\\ & & \times P_{2}\left(a_{2}|x_{2},\lambda_{1}\lambda_{2}\right)P_{3}\left(a_{3}|x_{3},\lambda_{2}\lambda_{3}\right) \end{array}$$
for all $a_{k}\in \left[o_{k}\right]\left(k=1,2,3\right)$, where
$${\left\{q_{k}\left(\lambda_{k}\right)\right\}}_{\lambda_{k}\in \left[n_{k}\right]},{\left\{P_{1}\left(a_{1}|x_{1},\lambda_{3}\lambda_{1}\right)\right\}}_{a_{1}\in \left[o_{1}\right]},{\left\{P_{2}\left(a_{2}|x_{2},\lambda_{1}\lambda_{2}\right)\right\}}_{a_{2}\in \left[o_{2}\right]},{\left\{P_{3}\left(a_{3}|x_{3},\lambda_{2}\lambda_{3}\right)\right\}}_{a_{3}\in \left[o_{3}\right]}$$
are PDs for all possible $x_{k},\lambda_{j}.$ Define
$$H_{1}^{\left(1\right)}=H_{1}^{\left(2\right)}=C^{n_{1}},H_{2}^{\left(2\right)}=H_{1}^{\left(3\right)}=C^{n_{2}},H_{2}^{\left(1\right)}=H_{2}^{\left(3\right)}=C^{n_{3}},$$
take their orthonormal bases $\{|\lambda_{3}\rangle {\}}_{\lambda_{3}=1}^{n_{3}}$, $\{|\lambda_{1}\rangle {\}}_{\lambda_{1}=1}^{n_{1}}$ and $\{|\lambda_{2}\rangle {\}}_{\lambda_{2}=1}^{n_{2}}$, respectively, and put
$$H^{\left(1\right)}=H_{2}^{\left(1\right)}⊗H_{1}^{\left(1\right)}=C^{n_{3}}⊗C^{n_{1}},H^{\left(2\right)}=H_{1}^{\left(2\right)}⊗H_{2}^{\left(2\right)}=C^{n_{1}}⊗C^{n_{2}},H^{\left(3\right)}=H_{1}^{\left(3\right)}⊗H_{2}^{\left(3\right)}=C^{n_{2}}⊗C^{n_{3}}$$
and choose separable states
$$\rho_{1,2}=\sum_{\lambda_{1}}q_{1}\left(\lambda_{1}\right)|\lambda_{1}\rangle \langle \lambda_{1}\left|⊗\right|\lambda_{1}\rangle \langle \lambda_{1}|\in D\left(H_{1}^{\left(1\right)}⊗H_{1}^{\left(2\right)}\right)=D\left(C^{n_{1}}⊗C^{n_{1}}\right),$$
$$\rho_{2,3}=\sum_{\lambda_{2}=1}^{n_{2}}q_{2}\left(\lambda_{2}\right)|\lambda_{2}\rangle \langle \lambda_{2}\left|⊗\right|\lambda_{2}\rangle \langle \lambda_{2}|\in D\left(H_{2}^{\left(2\right)}⊗H_{1}^{\left(3\right)}\right)=D\left(C^{n_{2}}⊗C^{n_{2}}\right),$$
$$\rho_{3,1}=\sum_{\lambda_{3}=1}^{n_{3}}q_{3}\left(\lambda_{3}\right)|\lambda_{3}\rangle \langle \lambda_{3}\left|⊗\right|\lambda_{3}\rangle \langle \lambda_{3}|\in D\left(H_{2}^{\left(3\right)}⊗H_{2}^{\left(1\right)}\right)=D\left(C^{n_{3}}⊗C^{n_{3}}\right),$$
then we obtain a triangle network TN with the network state
$$\begin{array}{ccc} \rho_{\mathrm{TN}} & = & \rho_{1,2}⊗\rho_{2,3}⊗\rho_{3,1}\\ & = & \sum_{\lambda_{1},\lambda_{2},\lambda_{3}}q_{1}\left(\lambda_{1}\right)q_{2}\left(\lambda_{2}\right)q_{3}\left(\lambda_{3}\right)|\lambda_{1}\rangle \langle \lambda_{1}\left|⊗\right|\lambda_{1}\rangle \langle \lambda_{1}\left|⊗\right|\lambda_{2}\rangle \langle \lambda_{2}\left|⊗\right|\lambda_{2}\rangle \langle \lambda_{2}\left|⊗\right|\lambda_{3}\rangle \langle \lambda_{3}\left|⊗\right|\lambda_{3}\rangle \langle \lambda_{3}|, \end{array}$$
inducing the measurement state
$$\begin{array}{c} \tilde{\rho_{\mathrm{TN}}}=\sum_{\lambda_{1},\lambda_{2},\lambda_{3}}q_{1}\left(\lambda_{1}\right)q_{2}\left(\lambda_{2}\right)q_{3}\left(\lambda_{3}\right)(|\lambda_{3}\rangle \langle \lambda_{3}\left|⊗\right|\lambda_{1}\rangle \langle \lambda_{1}|)⊗(|\lambda_{1}\rangle \langle \lambda_{1}\left|⊗\right|\lambda_{2}\rangle \langle \lambda_{2}|)⊗(|\lambda_{2}\rangle \langle \lambda_{2}\left|⊗\right|\lambda_{3}\rangle \langle \lambda_{3}|), \end{array}$$
in $D(H^{\left(1\right)}⊗H^{\left(2\right)}⊗H^{\left(3\right)})$. By defining positive operators:
$$M_{a_{1}|x_{1}}^{\left(1\right)}=\sum_{\lambda_{3}^{\prime }=1}^{n_{3}}\sum_{\lambda_{1}^{\prime }=1}^{n_{1}}P_{1}\left(a_{1}|x_{1},\lambda_{3}^{\prime }\lambda_{1}^{\prime }\right)|\lambda_{3}^{\prime }\lambda_{1}^{\prime }\rangle \langle \lambda_{3}^{\prime }\lambda_{1}^{\prime }|,$$
$$M_{a_{2}|x_{2}}^{\left(2\right)}=\sum_{\lambda_{1}^{\prime }=1}^{n_{1}}\sum_{\lambda_{2}^{\prime }=1}^{n_{2}}P_{2}\left(a_{2}|x_{2},\lambda_{1}^{\prime }\lambda_{2}^{\prime }\right)|\lambda_{1}^{\prime }\lambda_{2}^{\prime }\rangle \langle \lambda_{1}^{\prime }\lambda_{2}^{\prime }|,$$
$$M_{a_{3}|x_{3}}^{\left(3\right)}=\sum_{\lambda_{2}^{\prime }=1}^{n_{2}}\sum_{\lambda_{3}^{\prime }=1}^{n_{3}}P_{3}\left(a_{3}|x_{3},\lambda_{2}^{\prime }\lambda_{3}^{\prime }\right)|\lambda_{2}^{\prime }\lambda_{3}^{\prime }\rangle \langle \lambda_{2}^{\prime }\lambda_{3}^{\prime }|$$
on $H^{\left(1\right)},H^{\left(2\right)}$ and $H^{\left(3\right)}$, respectively, we obtain POVMs ${\left\{M_{a_{k}}^{\left(k\right)}\right\}}_{a_{k}=1}^{o_{k}}$ of system $H^{\left(k\right)}$ for each k=1,2,3. It is easy to check that
$$P\left(a_{1}a_{2}a_{3}|x_{1}x_{2}x_{3}\right)=\mathrm{tr}\left[\left({⊗}_{n=1}^{3}M_{a_{n}|x_{n}}^{\left(n\right)}\right)\tilde{\rho_{\mathrm{TN}}}\right],\;\forall a_{k}\in \left[o_{k}\right],x_{k}\in \left[m_{k}\right].$$
This shows that P is separable tri-quantum. The proof is completed. □

To discuss geometric and topological properties of C-trilocal and D-trilocal CTs, we have to put them into a topological space. A natural way is to consider the real Hilbert space $T(\Delta_{3})$ consisting of all correlation-type tensors [35] $P=〚P(a_{1}a_{2}a_{3}|x_{1}x_{2}x_{3})〛$ over $\Delta_{3}$ defined by functions $P:\Delta_{3}\to R$, in which the operations and inner products are given by
$$sP+tQ=〚sP\left(a_{1}a_{2}a_{3}|x_{1}x_{2}x_{3}\right)+tQ\left(a_{1}a_{2}a_{3}|x_{1}x_{2}x_{3}\right)〛,$$
$$\langle P|Q\rangle =\sum_{a_{i},x_{i}}P\left(a_{1}a_{2}a_{3}|x_{1}x_{2}x_{3}\right)Q\left(a_{1}a_{2}a_{3}|x_{1}x_{2}x_{3}\right)$$
for all s,t∈R and all elements P and Q of $T(\Delta_{3})$. The norm induced by the inner product reads
$$∥P∥={\left(\sum_{a_{i},x_{i}}{\left|P\left(a_{1}a_{2}a_{3}|x_{1}x_{2}x_{3}\right)\right|}^{2}\right)}^{\frac{1}{2}}$$
and then a sequence ${\left\{P_{n}\right\}}_{n=1}^{\infty }=\{〚P_{n}\left(a_{1}a_{2}a_{3}|x_{1}x_{2}x_{3}\right)〛{\}}_{n=1}^{\infty }$ in $T(\Delta_{3})$ is convergent (in norm) to $P=〚P(a_{1}a_{2}a_{3}|x_{1}x_{2}x_{3})〛$ if and only if
$$\lim_{n\to \infty }P_{n}\left(a_{1}a_{2}a_{3}|x_{1}x_{2}x_{3}\right)=P\left(a_{1}a_{2}a_{3}|x_{1}x_{2}x_{3}\right),\;\forall x_{i}\in \left[m_{i}\right],a_{i}\in \left[o_{i}\right]\left(i=1,2,3\right).$$
Thus, the set $\mathrm{CT}(\Delta_{3})$ of all CTs over $\Delta_{3}$ forms a compact convex set in $T(\Delta_{3})$. Since the hidden variables in a C-triLHVM or a D-triLHVM are assumed to be independent, the sets ${\mathrm{CT}}^{C-\mathrm{trilocal}}\left(\Delta_{3}\right)$ and ${\mathrm{CT}}^{D-\mathrm{trilocal}}\left(\Delta_{3}\right)$ are not necessarily convex. However, we have the following.

**Theorem** **2.**
**(Path-connectedness).**
*Both*

${\mathrm{CT}}^{C-\mathrm{trilocal}}\left(\Delta_{3}\right)$

*and*

${\mathrm{CT}}^{D-\mathrm{trilocal}}\left(\Delta_{3}\right)$

*are path-connected sets in the Hilbert space*

$T(\Delta_{3})$

*.*


**Proof.** Let $P=〚P(a_{1}a_{2}a_{3}|x_{1}x_{2}x_{3})〛$ and $Q=〚Q(a_{1}a_{2}a_{3}|x_{1}x_{2}x_{3})〛$ be any two elements of ${\mathrm{CT}}^{C-\mathrm{trilocal}}\left(\Delta_{3}\right)$. Then, P and Q have C-trLHVMs:
$$\begin{array}{c} P\left(a_{1}a_{2}a_{3}|x_{1}x_{2}x_{3}\right)=\int_{\Lambda }p_{1}\left(\lambda_{1}\right)p_{2}\left(\lambda_{2}\right)p_{3}\left(\lambda_{3}\right)P_{1}\left(a_{1}|x_{1},\lambda_{3}\lambda_{1}\right)P_{2}\left(a_{2}|x_{2},\lambda_{1}\lambda_{2}\right)P_{3}\left(a_{3}|x_{3},\lambda_{2}\lambda_{3}\right)d\mu \left(\lambda \right), \end{array}$$
and
$$\begin{array}{c} Q\left(a_{1}a_{2}a_{3}|x_{1}x_{2}x_{3}\right)=\int_{\Gamma }q_{1}\left(\xi_{1}\right)q_{2}\left(\xi_{2}\right)q_{3}\left(\xi_{3}\right)Q_{1}\left(a_{1}|x_{1},\xi_{3}\xi_{1}\right)Q_{2}\left(a_{2}|x_{2},\xi_{1}\xi_{2}\right)Q_{3}\left(a_{3}|x_{3},\xi_{2}\xi_{3}\right)d\gamma \left(\xi \right) \end{array}$$
for all possible $a_{1},a_{2},a_{3}$. Put
$$P_{0}\left(a_{1}a_{2}a_{3}|x_{1}x_{2}x_{3}\right)\equiv \frac{1}{o_{1}o_{2}o_{3}},P_{0}:=〚P_{0}\left(a_{1}a_{2}a_{3}|x_{1}x_{2}x_{3}\right)〛,$$
then $P_{0}$ is a D-trilocal (and then C-trilocal) CT over $\Delta_{3}.$ For every t∈[0,1/2], set
$$P_{1}^{t}\left(a_{1}|x_{1},\lambda_{3}\lambda_{1}\right)=\left(1-2t\right)P_{1}\left(a_{1}|x_{1},\lambda_{3}\lambda_{1}\right)+2t\frac{1}{o_{1}};$$
$$P_{2}^{t}\left(a_{2}|x_{2},\lambda_{1}\lambda_{2}\right)=\left(1-2t\right)P_{2}\left(a_{2}|x_{2},\lambda_{1}\lambda_{2}\right)+2t\frac{1}{o_{2}};$$
$$P_{3}^{t}\left(a_{3}|x_{3},\lambda_{2}\lambda_{3}\right)=\left(1-2t\right)P_{3}\left(a_{3}|x_{3},\lambda_{2}\lambda_{3}\right)+2t\frac{1}{o_{3}},$$
which are clearly PDs of $a_{1},a_{2}$ and $a_{3}$, respectively. Putting
$$\begin{array}{c} P^{t}\left(a_{1}a_{2}a_{3}|x_{1}x_{2}x_{3}\right)=\int_{\Lambda }q_{1}\left(\lambda_{1}\right)q_{2}\left(\lambda_{2}\right)q_{3}\left(\lambda_{3}\right)P_{1}^{t}\left(a_{1}|x_{1},\lambda_{3}\lambda_{1}\right)P_{2}^{t}\left(a_{2}|x_{2},\lambda_{1}\lambda_{2}\right)P_{3}^{t}\left(a_{3}|x_{3},\lambda_{2}\lambda_{3}\right)d\mu \left(\lambda \right), \end{array}$$
then $P\left(t\right):=〚P^{t}\left(a_{1}a_{2}a_{3}|x_{1}x_{2}x_{3}\right)〛$ is a C-trilocal CT over $\Delta_{3}$ for every t∈[0,1/2] with P(0)=P and $P\left(1/2\right)=P_{0}$. Obviously, the map t↦P(t) from [0,1/2] into ${\mathrm{PT}}^{C-\mathrm{trilocal}}\left(\Omega_{3}\right)$ is continuous.Similarly, for every t∈[1/2,1], set
$$Q_{1}^{t}\left(a_{1}|x_{1},\xi_{3}\xi_{1}\right)=\left(2t-1\right)Q_{1}\left(a_{1}|x_{1},\xi_{3}\xi_{1}\right)+2\left(1-t\right)\frac{1}{o_{1}};$$
$$Q_{2}^{t}\left(a_{2}|x_{2},\xi_{1}\xi_{2}\right)=\left(2t-1\right)Q_{2}\left(a_{2}|x_{2},\xi_{1}\xi_{2}\right)+2\left(1-t\right)\frac{1}{o_{2}};$$
$$Q_{3}^{t}\left(a_{3}|x_{3},\xi_{2}\xi_{3}\right)=\left(2t-1\right)Q_{3}\left(a_{3}|x_{3},\xi_{2}\xi_{3}\right)+2\left(1-t\right)\frac{1}{o_{3}},$$
which are clearly PDs of $a_{1},a_{2}$ and $a_{3}$, respectively. Putting
$$\begin{array}{c} Q^{t}\left(a_{1}a_{2}a_{3}|x_{1}x_{2}x_{3}\right)=\int_{\Gamma }q_{1}\left(\xi_{1}\right)q_{2}\left(\xi_{2}\right)q_{3}\left(\xi_{3}\right)Q_{1}^{t}\left(a_{1}|x_{1},\xi_{3}\xi_{1}\right)Q_{2}^{t}\left(a_{2}|x_{2},\xi_{1}\xi_{2}\right)Q_{3}^{t}\left(a_{3}|x_{3},\xi_{2}\xi_{3}\right)d\gamma \left(\xi \right), \end{array}$$
then $Q\left(t\right):=〚Q^{t}\left(a_{1}a_{2}a_{3}|x_{1}x_{2}x_{3}\right)〛$ is a C-trilocal CT over $\Delta_{3}$ for every t∈[1/2,1] with $Q\left(1/2\right)=P_{0}$ and Q(1)=Q. Obviously, the map t↦Q(t) from [1/2,1] into ${\mathrm{PT}}^{C-\mathrm{trilocal}}\left(\Delta_{3}\right)$ is continuous.Define a mapping $f:\left[0,1\right]\to {\mathrm{CT}}^{C-\mathrm{trilocal}}\left(\Delta_{3}\right)$ by
$$f\left(t\right)=\left\{\begin{array}{cc} P(t), & t\in [0,1/2];\\ Q(t), & t\in (1/2,1], \end{array}\right.$$
then *f* is continuous everywhere and and then induces a path in ${\mathrm{CT}}^{C-\mathrm{trilocal}}\left(\Delta_{3}\right)$, connecting P and Q. This shows that ${\mathrm{CT}}^{C-\mathrm{trilocal}}\left(\Delta_{3}\right)$ is path-connected. Similarly, ${\mathrm{CT}}^{D-\mathrm{trilocal}}\left(\Delta_{3}\right)$ is also path-connected. The proof is completed. □

For k=1,2,3, taking a CT $E_{k}=〚E_{k}\left(a_{k}|x_{k}\right)〛$ over $\left[o_{k}\right]\times \left[m_{k}\right]$ and defining
$$S_{1}\left(a_{1}a_{2}a_{3}|x_{1}x_{2}x_{3}\right)=E_{1}\left(a_{1}|x_{1}\right)\times \frac{1}{o_{2}}\times \frac{1}{o_{3}},$$
$$S_{2}\left(a_{1}a_{2}a_{3}|x_{1}x_{2}x_{3}\right)=\frac{1}{o_{1}}\times E_{2}\left(a_{2}|x_{2}\right)\times \frac{1}{o_{3}},$$
$$S_{3}\left(a_{1}a_{2}a_{3}|x_{1}x_{2}x_{3}\right)=\frac{1}{o_{1}}\times \frac{1}{o_{2}}\times E_{3}\left(a_{3}|x_{3}\right),$$
we obtain three CTs $S_{k}:=〚S_{k}\left(a_{1}a_{2}a_{3}|x_{1}x_{2}x_{3}\right)〛$ over $\Delta_{3}$ with
$$\sum_{a_{i}\left(i\neq k\right)}S_{k}\left(a_{1}a_{2}a_{3}|x_{1}x_{2}x_{3}\right)=E_{k}\left(a_{k}|x_{k}\right)$$
for k=1,2,3. Clearly, $S_{k}$ is D-trilocal and then C-trilocal CT over $\Delta_{3}$ for each *k*. Put
$${\mathrm{CT}}_{E_{k}}^{C-\mathrm{trilocal}}\left(\Delta_{3}\right)=\left\{P\in {\mathrm{CT}}^{C-\mathrm{trilocal}}\left(\Delta_{3}\right):P_{k}=E_{k}\right\},$$
where
$$P_{k}\left(a_{k}|x_{k}\right):=\sum_{a_{i}\left(i\neq k\right)}P\left(a_{1}a_{2}a_{3}|x_{1}x_{2}x_{3}\right)$$
denotes the marginal distribution of $P(a_{1}a_{2}a_{3}|x_{1}x_{2}x_{3})$ on the *k*-th node.

**Theorem** **3.**
**(Partial star-convexity).**
*The set*

${\mathrm{CT}}_{E_{k}}^{C-\mathrm{trilocal}}\left(\Delta_{3}\right)$

*is star-convex with a sun*

$S_{k}$

*for each*

k=1,2,3

*, i.e.,*

(30)
$$tS_{k}+\left(1-t\right){\mathrm{CT}}_{E_{k}}^{C-\mathrm{trilocal}}\left(\Delta_{3}\right)\subset {\mathrm{CT}}_{E_{k}}^{C-\mathrm{trilocal}}\left(\Delta_{3}\right),\;\forall t\in \left[0,1\right].$$



**Proof.** Let $P=〚P\left(a_{1}a_{2}a_{3}|x_{1}x_{2}x_{3}\right)〛\in {\mathrm{CT}}_{E_{1}}^{C-\mathrm{trilocal}}\left(\Delta_{3}\right)$. Then, P has a C-triLHVM:
(31)$$\begin{array}{ccc} P(a_{1}a_{2}a_{3}|x_{1}x_{2}x_{3}) & = & \int_{\Lambda }p_{1}\left(\lambda_{1}\right)p_{2}\left(\lambda_{2}\right)p_{3}\left(\lambda_{3}\right)P_{1}\left(a_{1}|x_{1},\lambda_{3}\lambda_{1}\right)\\ & & \times P_{2}\left(a_{2}|x_{2},\lambda_{1}\lambda_{2}\right)P_{3}\left(a_{3}|x_{3},\lambda_{2}\lambda_{3}\right)d\mu \left(\lambda \right), \end{array}$$
where $\left(\Lambda ,\Omega ,\mu \right)=\left(\Lambda_{1}\times \Lambda_{2}\times \Lambda_{3},\Omega_{1}\times \Omega_{2}\times \Omega_{3},\mu_{1}\times \mu_{2}\times \mu_{3}\right)$ is a product measure space with $\lambda =(\lambda_{1},\lambda_{2},\lambda_{3})$. Thus,
(32)$$\begin{array}{ccc} & & E\left(a_{1}|x_{1}\right)=P_{1}\left(a_{1}|x_{1}\right):=\sum_{a_{2},a_{3}}P\left(a_{1}a_{2}a_{3}|x_{1}x_{2}x_{3}\right)\\ & & =\int_{\Lambda_{1}\times \Lambda_{3}}p_{1}\left(\lambda_{1}\right)p_{3}\left(\lambda_{3}\right)P_{1}\left(a_{1}|x_{1},\lambda_{3}\lambda_{1}\right)d\mu_{1}\left(\lambda_{1}\right)d\mu_{3}\left(\lambda_{3}\right). \end{array}$$
Put P({0,1})={∅,{0},{1},{0,1}}, which is a σ-algebra on {0,1}, and set
$$\Lambda_{2}=\Lambda \times \left\{0,1\right\},\Omega_{2}^{\prime }=\Omega_{2}\times P\left(\left\{0,1\right\}\right),\lambda_{2}^{\prime }=\left(\lambda_{2},s\right),\mu_{2}^{\prime }=\mu_{2}\times c,$$
where *c* denotes the counting measure on {0,1}. Then, we obtain a product measure space
$$(\Lambda_{1}\times \Lambda_{2}^{\prime }\times \Lambda_{3},\Omega_{1}\times \Omega_{2}^{\prime }\times \Omega_{3},\mu_{1}\times \mu_{2}^{\prime }\times \mu_{3}).$$
For every t∈[0,1] and every $\lambda_{2}^{\prime }=\left(\lambda_{2},s\right)$, set
$$f\left(\lambda_{2}^{\prime }\right)=\left\{\begin{array}{cc} p_{2}\left(\lambda_{2}\right)\left(1-t\right), & s=0;\\ p_{2}\left(\lambda_{2}\right)t, & s=1, \end{array}\right.$$
which is a DF of $\lambda_{2}^{\prime }$; define
$$P_{2}\left(a_{2}|x_{2},\lambda_{1}\lambda_{2}^{\prime }\right)=\left\{\begin{array}{cc} \frac{1}{o_{2}}, & s=0;\\ P_{B}\left(a_{2}|x_{2},\lambda_{1}\lambda_{2}\right), & s=1, \end{array}\right.$$
$$P_{3}\left(a_{3}|x_{3},\lambda_{2}^{\prime }\lambda_{3}\right)=\left\{\begin{array}{cc} \frac{1}{o_{3}}, & s=0;\\ P_{C}\left(a_{3}|x_{3},\lambda_{2}\lambda_{3}\right), & s=1, \end{array}\right.$$
which are PDs of $a_{2}$ and $a_{3}$, respectively. For all $x_{1},x_{2},x_{3},a_{1},a_{2},a_{3}$, we see from (Equation 32) and (Equation 31) that
$$\begin{array}{ccc} & & \int_{\Lambda_{1}\times \Lambda_{2}^{\prime }\times \Lambda_{3}}p_{1}\left(\lambda_{1}\right)f\left(\lambda_{2},s\right)p_{3}\left(\lambda_{3}\right)P_{1}\left(a_{1}|x_{1},\lambda_{3}\lambda_{1}\right)\\ & & \times P_{2}\left(a_{2}|x_{2},\lambda_{1}\lambda_{2}^{\prime }\right)P_{3}\left(a_{3}|x_{3},\lambda_{2}^{\prime }\lambda_{3}\right)d\mu_{1}\left(\lambda_{1}\right)d\mu_{2}^{\prime }\left(\lambda_{2}^{\prime }\right)d\mu_{3}\left(\lambda_{3}\right)\\ & = & \int_{\Lambda_{1}\times \Lambda_{2}\times \Lambda_{3}}p_{1}\left(\lambda_{1}\right)p_{2}\left(\lambda_{2}\right)p_{3}\left(\lambda_{3}\right)\left(1-t\right)P_{1}\left(a_{1}|x_{1},\lambda_{3}\lambda_{1}\right)\\ & & \times \frac{1}{o_{2}}\frac{1}{o_{3}}d\mu_{1}\left(\lambda_{1}\right)d\mu_{2}\left(\lambda_{2}\right)d\mu_{3}\left(\lambda_{3}\right)\\ & & +\int_{\Lambda_{1}\times \Lambda_{2}\times \Lambda_{3}}p_{1}\left(\lambda_{1}\right)p_{2}\left(\lambda_{2}\right)p_{3}\left(\lambda_{3}\right)tP_{1}\left(a_{1}|x_{1},\lambda_{3}\lambda_{1}\right)\\ & & \times P_{2}\left(a_{2}|x_{2},\lambda_{1}\lambda_{2}\right)P_{3}\left(a_{3}|x_{3},\lambda_{2}\lambda_{3}\right)d\mu_{1}\left(\lambda_{1}\right)d\mu_{2}\left(\lambda_{2}\right)d\mu_{3}\left(\lambda_{3}\right)\\ & = & \left(1-t\right)S\left(a_{1}a_{2}a_{3}|x_{1}x_{2}x_{3}\right)+tP\left(a_{1}a_{2}a_{3}|x_{1}x_{2}x_{3}\right). \end{array}$$This shows that $\left(1-t\right)S_{1}+tP$ is C-trilocal with $S_{1}=E_{1}$ and then an element of ${\mathrm{CT}}_{E_{1}}^{C-\mathrm{trilocal}}\left(\Delta_{3}\right)$. Thus,
$$tS_{1}+\left(1-t\right){\mathrm{CT}}_{E_{1}}^{C-\mathrm{trilocal}}\left(\Delta_{3}\right)\subset {\mathrm{CT}}_{E_{1}}^{C-\mathrm{trilocal}}\left(\Delta_{3}\right)$$
for all t∈[0,1]. That is, ${\mathrm{CT}}_{E_{1}}^{C-\mathrm{trilocal}}\left(\Delta_{3}\right)$ is star-convex with a sun $S_{1}$. Similarly, ${\mathrm{CT}}_{E_{k}}^{C-\mathrm{trilocal}}\left(\Delta_{3}\right)$ is star-convex with a sun $S_{k}$ for k=2,3. The proof is completed. □

**Remark** **2.**
*Let p=〚p(i,j,k)〛 be a C-trilocal PT over a finite set I×J×K with a C-triLHVM:*

$$p\left(i,j,k\right)=\int_{\Lambda }q_{1}\left(\lambda_{1}\right)q_{2}\left(\lambda_{2}\right)q_{3}\left(\lambda_{3}\right)P_{1}\left(i|\lambda_{3}\lambda_{1}\right)P_{2}\left(j|\lambda_{1}\lambda_{2}\right)P_{3}\left(k|\lambda_{2}\lambda_{3}\right)d\mu \left(\lambda \right),$$

*where $q_{j}\left(\lambda_{j}\right)$ is a DF of $\lambda_{j}$, $P_{1}\left(i|\lambda_{3}\lambda_{1}\right),P_{2}\left(j|\lambda_{1}\lambda_{2}\right),P_{3}\left(k|\lambda_{2}\lambda_{3}\right)$ are PDs of $\lambda_{j},i,j$ and k, respectively. Suppose that ${\left\{P_{i}\left(a_{1}|x_{1}\right)\right\}}_{a_{1}\in \left[o_{1}\right]},{\left\{P_{j}\left(a_{2}|x_{2}\right)\right\}}_{a_{2}\in \left[o_{2}\right]}$ and ${\left\{P_{k}\left(a_{3}|x_{3}\right)\right\}}_{a_{3}\in \left[o_{3}\right]}$ are PDs of $a_{1},a_{2}$ and $a_{3}$, respectively, Thus, the CT P defined by*

(33)
$$P\left(a_{1}a_{2}a_{3}|x_{1}x_{2}x_{3}\right)=\sum_{i,j,k}p\left(i,j,k\right)P_{i}\left(a_{1}|x_{1}\right)P_{j}\left(a_{2}|x_{2}\right)P_{k}\left(a_{3}|x_{3}\right)$$

*can be written as*

$$\begin{array}{ccc} P(a_{1}a_{2}a_{3}|x_{1}x_{2}x_{3}) & = & \sum_{i,j,k}p\left(i,j,k\right)P_{i}\left(a_{1}|x_{1}\right)P_{j}\left(a_{2}|x_{2}\right)P_{k}\left(a_{3}|x_{3}\right)\\ & = & \int_{\Lambda }q_{1}\left(\lambda_{1}\right)q_{2}\left(\lambda_{2}\right)q_{3}\left(\lambda_{3}\right)P_{1}\left(a_{1}|x_{1},\lambda_{3}\lambda_{1}\right)\\ & & \times P_{2}\left(a_{2}|x_{2},\lambda_{1}\lambda_{2}\right)P_{3}\left(a_{3}|x_{3},\lambda_{2}\lambda_{3}\right)d\mu \left(\lambda \right), \end{array}$$

*where*

$$P_{1}\left(a_{1}|x_{1},\lambda_{3}\lambda_{1}\right)=\sum_{i\in I}P_{1}\left(i|\lambda_{3}\lambda_{1}\right)P_{i}\left(a_{1}|x_{1}\right),$$


$$P_{2}\left(a_{2}|x_{2},\lambda_{1}\lambda_{2}\right)=\sum_{j\in J}P_{2}\left(j|\lambda_{1}\lambda_{2}\right)P_{j}\left(a_{2}|x_{2}\right),$$


$$P_{3}\left(a_{3}|x_{3},\lambda_{2}\lambda_{3}\right)=\sum_{k\in K}P_{3}\left(k|\lambda_{2}\lambda_{3}\right)P_{k}\left(a_{3}|x_{3}\right),$$


*which are PDs of $a_{1},a_{2}$ and $a_{3}$, respectively. Thus, P is a C-trilocal CT over $\Delta_{3}$. In particular, when*

$$N_{i}=o_{i}^{m_{i}}\left(i=1,2,3\right),\Gamma_{3}=\left[N_{1}\right]\times \left[N_{2}\right]\times \left[N_{3}\right],p=〚p\left(i,j,k\right)〛\in {\mathrm{PT}}^{C-\mathrm{trilocal}}\left(\Gamma_{3}\right),$$


*we obtain that $P:=\sum_{i,j,k}p\left(i,j,k\right)D_{ijk}$ is a C-trilocal CT over $\Delta_{3}$, where*

$$D_{ijk}=〚D_{ijk}\left(a_{1}a_{2}a_{3}|x_{1}x_{2}x_{3}\right)〛=〚\delta_{a_{1},J_{i}\left(x_{1}\right)}\delta_{a_{2},K_{j}\left(x_{2}\right)}\delta_{a_{3},L_{k}\left(x_{3}\right)}〛,$$


*in which*

$$\left\{J_{1},J_{2},\ldots ,J_{N_{1}}\right\}=\left\{J|J:\left[m_{1}\right]\to \left[o_{1}\right]\right\},$$


$$\left\{K_{1},K_{2},\ldots ,K_{N_{2}}\right\}=\left\{K|K:\left[m_{2}\right]\to \left[o_{2}\right]\right\},$$


$$\left\{L_{1},L_{2},\ldots ,L_{N_{3}}\right\}=\left\{L|L:\left[m_{3}\right]\to \left[o_{3}\right]\right\}.$$


*Clearly, $D_{ijk}$’s are D-trilocal CTs over $\Delta_{3}$. This shows that*

(34)
$${\mathrm{CT}}^{C-\mathrm{trilocal}}\left(\Delta_{3}\right)⊃\left\{\sum_{i,j,k}p\left(i,j,k\right)D_{ijk}:\;p=〚p\left(i,j,k\right)〛\in {\mathrm{PT}}^{C-\mathrm{trilocal}}\left(\Gamma_{3}\right)\right\}.$$


*Similarly,*

(35)
$${\mathrm{CT}}^{D-\mathrm{trilocal}}\left(\Delta_{3}\right)⊃\left\{\sum_{i,j,k}p\left(i,j,k\right)D_{ijk}:\;p=〚p\left(i,j,k\right)〛\in {\mathrm{PT}}^{D-\mathrm{trilocal}}\left(\Gamma_{3}\right)\right\}.$$



Next, we aim to show that Equations (Equation 34) and (Equation 35) are indeed equalities. To do this, we recall that an m×n function matrix $B\left(\lambda \right)=[b_{ij}\left(\lambda \right)]$ on Λ is said to be row-stochastic (RS) means that, for each λ∈Λ, $b_{ij}\left(\lambda \right)\geq 0$ for all i,j and $\sum_{j=1}^{n}b_{ij}\left(\lambda \right)=1$ for all i∈[m]. It is clear that every m×n{0,1}-row statistics matrix $T=[T_{ij}]$ corresponds uniquely a mapping F:[m]→[n] so that $T_{ij}=\delta_{j,F(i)}$. Thus, the sets of all {0,1}-row-stochastic matrices of orders $m_{1}\times o_{1}$, $m_{2}\times o_{2}$, and $m_{3}\times o_{3}$ can be written as
$$RSM_{m_{1}\times o_{1}}^{\left(0,1\right)}=\left\{R_{i}:={\left[\delta_{a_{1},J_{i}\left(x_{1}\right)}\right]}_{x_{1},a_{1}}:i=1,2,\ldots ,N_{1}\right\},$$
$$RSM_{m_{2}\times o_{2}}^{\left(0,1\right)}=\left\{K_{j}:={\left[\delta_{a_{2},K_{j}\left(x_{2}\right)}\right]}_{x_{2},a_{2}}:j=1,2,\ldots ,N_{2}\right\},$$
$$RSM_{m_{3}\times o_{3}}^{\left(0,1\right)}=\left\{L_{k}:={\left[\delta_{a_{3},L_{k}\left(x_{3}\right)}\right]}_{x_{3},a_{3}}:k=1,2,\ldots ,N_{3}\right\},$$
respectively.

**Lemma** **1**([36]). *Let (Λ,Ω,μ) be a measure space. Then, every m×n function RS matrix $B\left(\lambda \right)=[b_{ij}\left(\lambda \right)]$ on Λ whose entries are Ω-measurable on Λ can be written as a convex combination of all {0,1}-RS matrices $R_{k}$’s:*
(36)$$B\left(\lambda \right)=\sum_{k=1}^{n^{m}}\alpha_{k}\left(\lambda \right)R_{k},\;\forall \lambda \in \Lambda ,$$*where $\alpha_{k}\left(k=1,2,\ldots ,n^{m}\right)$ are all nonnegative and Ω-measurable functions on* Λ.

Using ([35] Theorem 5.1) implies that
(37)$${\mathrm{CT}}^{\mathrm{Bell}-\mathrm{local}}\left(\Delta_{3}\right)=\left\{\sum_{i,j,k}p\left(i,j,k\right)D_{ijk}:\;p=〚p\left(i,j,k\right)〛\in \mathrm{PT}\left(\Gamma_{3}\right)\right\},$$
where $\mathrm{PT}(\Gamma_{3})$ denotes the set of all PTs over $\Gamma_{3}$. Based this lemma, we can show the following conclusion, which say that a CT over $\Delta_{3}$ is C-trilocal (resp. D-trilocal) if and only if it can be written as a convex combination of local deterministic CTs $D_{ijk}$’s with C-trilocal (resp. D-trilocal) coefficients.

**Theorem** **4.**

(38)
$${\mathrm{CT}}^{C-\mathrm{trilocal}}\left(\Delta_{3}\right)=\left\{\sum_{i,j,k}p\left(i,j,k\right)D_{ijk}:\;p=〚p\left(i,j,k\right)〛\in {\mathrm{PT}}^{C-\mathrm{trilocal}}\left(\Gamma_{3}\right)\right\},$$


(39)
$${\mathrm{CT}}^{D-\mathrm{trilocal}}\left(\Delta_{3}\right)=\left\{\sum_{i,j,k}p\left(i,j,k\right)D_{ijk}:\;p=〚p\left(i,j,k\right)〛\in {\mathrm{PT}}^{D-\mathrm{trilocal}}\left(\Gamma_{3}\right)\right\}.$$



**Proof.** Suppose that P is C-trilocal; then, it has a C-triLHVM (Equation 20). Since matrices
$$M\left(\lambda_{3},\lambda_{1}\right):={\left[P_{1}\left(a_{1}|x_{1},\lambda_{3}\lambda_{1}\right)\right]}_{x_{1},a_{1}}\in R^{m_{1}\times o_{1}},$$
$$M\left(\lambda_{1},\lambda_{2}\right):={\left[P_{2}\left(a_{2}|x_{2},\lambda_{1}\lambda_{2}\right)\right]}_{x_{2},a_{2}}\in R^{m_{2}\times o_{2}},$$
$$M\left(\lambda_{2},\lambda_{3}\right):={\left[P_{3}\left(a_{2}|x_{3},\lambda_{2}\lambda_{3}\right)\right]}_{x_{3},a_{3}}\in R^{m_{3}\times o_{3}}$$
are row-stochastic with measurable entries, we see from Lemma 1 that they have the following decompositions:
(40)$$P_{1}\left(a_{1}|x_{1},\lambda_{3}\lambda_{1}\right)=\sum_{i=1}^{N_{1}}P_{1}\left(i|\lambda_{3}\lambda_{1}\right)\delta_{a_{1},J_{i}\left(x_{1}\right)},$$
(41)$$P_{2}\left(a_{2}|x_{2},\lambda_{1}\lambda_{2}\right)=\sum_{j=1}^{N_{2}}P_{2}\left(j|\lambda_{1}\lambda_{2}\right)\delta_{a_{2},K_{j}\left(x_{2}\right)},$$
(42)$$P_{3}\left(a_{3}|x_{3},\lambda_{2}\lambda_{3}\right)=\sum_{k=1}^{N_{3}}P_{3}\left(k|\lambda_{2}\lambda_{3}\right)\delta_{a_{3},L_{k}\left(x_{3}\right)},$$
where $P_{1}\left(i|\lambda_{3}\lambda_{1}\right),P_{2}\left(j|\lambda_{1}\lambda_{2}\right)$ and $P_{3}\left(k|\lambda_{2}\lambda_{3}\right)$ are PDs of i,j and *k*, respectively, and measurable w.r.t. $\left(\lambda_{3},\lambda_{1}\right),\left(\lambda_{1},\lambda_{2}\right)$ and $\left(\lambda_{2},\lambda_{3}\right)$, respectively. Hence,
(43)$$P\left(a_{1}a_{2}a_{3}|x_{1}x_{2}x_{3}\right)=\sum_{i,j,k}p\left(i,j,k\right)\delta_{a_{1},J_{i}\left(x_{1}\right)}\delta_{a_{2},K_{j}\left(x_{2}\right)}\delta_{a_{3},L_{k}\left(x_{3}\right)},$$
where
(44)$$p\left(i,j,k\right)=\int_{\Lambda }q_{1}\left(\lambda_{1}\right)q_{2}\left(\lambda_{2}\right)q_{3}\left(\lambda_{3}\right)P_{1}\left(i|\lambda_{3}\lambda_{1}\right)P_{2}\left(j|\lambda_{1}\lambda_{2}\right)P_{3}\left(k|\lambda_{2}\lambda_{3}\right)d\mu \left(\lambda \right),$$
which forms a C-trilocal PT p=〚p(i,j,k)〛 over $\Gamma_{3}$, satisfying
$$P=\sum_{i,j,k}p\left(i,j,k\right)D_{ijk}.$$Conversely, if p=〚p(i,j,k)〛 is a C-trilocal PT over $\Gamma_{3}$, then it has a C-triLVHM (Equation 44), and so the CT $P=〚P(a_{1}a_{2}a_{3}|x_{1}x_{2}x_{3})〛$ defined by (Equation 43) has a C-triLHVM (Equation 20) in light of (Equation 40)–(Equation 42). Thus, P becomes a C-trilocal CT over $\Delta_{3}$ and Equation (Equation 38) follows. Similarly, (Equation 39) is also valid. The proof is completed. □

Theorem 4 implies that both D-trilocal and C-trilocal CTs over $\Delta_{3}$ are Bell local. It also yields that every C-trilocal CT P over $\Delta_{3}$ can be written as a convex combination (Equation 43) of the deterministic D-bilocal CTs $D_{ijk}$ over $\Delta_{3}$.

**Corollary** **1.**

(45)
$${\mathrm{CT}}^{C-\mathrm{trilocal}}\left(\Delta_{3}\right)\subset \mathrm{conv}\left({\mathrm{CT}}^{D-\mathrm{bilocal}}\left(\Delta_{3}\right)\right)={\mathrm{CT}}^{\mathrm{Bell}-\mathrm{local}}\left(\Delta_{3}\right).$$



Let ${\mathrm{CT}}_{R}^{C-\mathrm{trilocal}}\left(\Delta_{3}\right)$ be the set of all C-trilocal CTs over $\Delta_{3}$ with C-triLHVMs given by three-hold Riemann integrals over a product region $\Lambda_{1}\times \Lambda_{2}\times \Lambda_{3}$.

**Theorem** **5.**

(46)
$${\mathrm{CT}}^{D-\mathrm{trilocal}}\left(\Delta_{3}\right)\subset {\mathrm{CT}}_{R}^{C-\mathrm{trilocal}}\left(\Delta_{3}\right)\subset \bar{{\mathrm{CT}}^{D-\mathrm{trilocal}}\left(\Delta_{3}\right)},$$

*where $\bar{{\mathrm{CT}}^{D-\mathrm{trilocal}}\left(\Delta_{3}\right)}$ denotes the closure of ${\mathrm{CT}}^{D-\mathrm{trilocal}}\left(\Delta_{3}\right)$ in the Hilbert space $T(\Delta_{3})$.*


**Proof.** The second inclusion can be checked in a way similar to the proof of Proposition 6. To check the first inclusion, we let $P\in {\mathrm{CT}}^{D-\mathrm{trilocal}}\left(\Delta_{3}\right).$ Then, it can be written as (Equation 21):
$$\begin{array}{ccc} P(a_{1}a_{2}a_{3}|x_{1}x_{2}x_{3}) & = & \sum_{\lambda_{1}=1}^{n_{1}}\sum_{\lambda_{2}=1}^{n_{2}}\sum_{\lambda_{3}=1}^{n_{3}}q_{1}\left(\lambda_{1}\right)q_{2}\left(\lambda_{2}\right)q_{3}\left(\lambda_{3}\right)P_{1}\left(a_{1}|x_{1},\lambda_{3}\lambda_{1}\right)\\ & & \times P_{2}\left(a_{2}|x_{2},\lambda_{1}\lambda_{2}\right)P_{3}\left(a_{3}|x_{3},\lambda_{2}\lambda_{3}\right) \end{array}$$
for all $x_{k}\in \left[m_{k}\right],a_{k}\in \left[o_{k}\right]\left(k=1,2,3\right)$, where
$$q_{k}\left(\lambda_{k}\right),P_{1}\left(a_{1}|x_{1},\lambda_{3}\lambda_{1}\right),P_{2}\left(a_{2}|x_{2},\lambda_{1}\lambda_{2}\right),P_{3}\left(a_{3}|x_{3},\lambda_{2}\lambda_{3}\right)$$
are PDs of $\lambda_{k},a_{1},a_{2},a_{3}$, respectively. By using the characteristic function of a set *S*:
$$\chi_{S}\left(x\right)=\left\{\begin{array}{cc} 1, & x\in S;\\ 0, & x\notin S, \end{array}\right.$$
we define functions:
$$p_{k}\left(t_{k}\right)=\sum_{\lambda_{k}}q_{k}\left(\lambda_{k}\right)\chi_{\left[\lambda_{k}-1,\lambda_{k}\right)}\left(t_{k}\right)\left(\forall t_{k}\in \left[0,n_{k}\right)\right),\;p_{k}\left(n_{k}\right)=0,k=1,2,3,$$
$$Q_{1}\left(a_{1}|x_{1},t_{3}t_{1}\right)=\sum_{\lambda_{3},\lambda_{1}}P_{1}\left(a_{1}|x_{1},\lambda_{3}\lambda_{1}\right)\chi_{\left[\lambda_{1}-1,\lambda_{1}\right)\times \left[\lambda_{3}-1,\lambda_{3}\right)}\left(t_{1},t_{3}\right)$$
if $\left(t_{1},t_{3}\right)\in \left[0,n_{1}\right)\times \left[0,n_{3}\right);$$Q_{1}\left(a_{1}|x_{1},t_{3}t_{1}\right)=\frac{1}{o_{1}}$, otherwise;
$$Q_{2}\left(a_{2}|x_{2},t_{1}t_{2}\right)=\sum_{\lambda_{1},\lambda_{2}}P_{2}\left(a_{2}|x_{2},\lambda_{1}\lambda_{2}\right)\chi_{\left[\lambda_{1}-1,\lambda_{1}\right)\times \left[\lambda_{2}-1,\lambda_{2}\right)}\left(t_{1},t_{2}\right)$$
if $\left(t_{1},t_{2}\right)\in \left[0,n_{1}\right)\times \left[0,n_{2}\right);$$Q_{2}\left(a_{2}|x_{2},t_{1}t_{2}\right)=\frac{1}{o_{2}}$, otherwise;
$$Q_{3}\left(a_{3}|x_{3},t_{2}t_{3}\right)=\sum_{\lambda_{2},\lambda_{3}}P_{3}\left(a_{3}|x_{3},\lambda_{2}\lambda_{3}\right)\chi_{\left[\lambda_{2}-1,\lambda_{2}\right)\times \left[\lambda_{3}-1,\lambda_{3}\right)}\left(t_{2},t_{3}\right)$$
if $\left(t_{2},t_{3}\right)\in \left[0,n_{2}\right)\times \left[0,n_{3}\right);$$Q_{3}\left(a_{3}|x_{3},t_{2}t_{3}\right)=\frac{1}{o_{3}}$, otherwise. Clearly, $p_{k}\left(t_{k}\right)$ is a DF of $t_{k}\in \left[0,n_{k}\right]\left(k=1,2,3\right)$, $Q_{1}\left(a_{1}|x_{1},t_{3}t_{1}\right)$, $Q_{2}\left(a_{2}|x_{2},t_{1}t_{2}\right)$ and $Q_{3}\left(a_{3}|x_{3},t_{2}t_{3}\right)$ are PDs of $a_{1},a_{2}$ and $a_{3}$, respectively, for all $x_{k}\in \left[m_{k}\right]$ and all $t_{k}\in \left[0,n_{k}\right]$. It is easy to check that
$$\begin{array}{ccc} P(a_{1}a_{2}a_{3}|x_{1}x_{2}x_{3}) & = & \int_{0}^{n_{1}}\int_{0}^{n_{2}}\int_{0}^{n_{3}}p_{1}\left(t_{1}\right)p_{2}\left(t_{2}\right)p_{3}\left(t_{3}\right)Q_{1}\left(a_{1}|x_{1},t_{3}t_{1}\right)\\ & & \times Q_{2}\left(a_{2}|x_{2},t_{1}t_{2}\right)Q_{3}\left(a_{3}|x_{3},t_{2}t_{3}\right)dt_{1}dt_{2}dt_{3} \end{array}$$
for all possible $x_{i},a_{i}$. Thus, $P\in {\mathrm{CT}}_{R}^{C-\mathrm{trilocal}}\left(\Delta_{3}\right).$ This completes the proof. □

## 4. Conclusions and Questions

When a triangle network is locally measured one run or many runs, a probability tensor (PT) $P=〚P(a_{1}a_{2}a_{3})〛$ over $\Omega_{3}$ or a correlation tensor (CT) $P=〚P(a_{1}a_{2}a_{3}|x_{1}x_{2}x_{3})〛$ over $\Delta_{3}$ is obtained. In this work, we have introduced and discussed C-trilocality and D-trilocality of PTs and CTs according to their descriptions of continuous (integral) and discrete (sum) trilocal hidden variable models (C-triLHVMs and D-triLHVMs). We named that a PT (or CT) P is C-trilocal (resp. D-trilocal) if it can be described by a C-triLHVM (resp. D-triLHVM). With these definitions, the following conclusions have been proved:

(1) A PT (resp. CT) is D-trilocal if and only if it can be realized in a triangle network by three shared separable states and a local POVM (resp. a set of local POVMs);

(2) A CT is C-trilocal (resp. D-trilocal) if and only if it can be written as a convex combination of the product deterministic CTs with a C-trilocal (resp. D-trilocal) PT as coefficient tensor;

(3) When one of the shared states $\rho_{i,j}$ in the triangle network is Bell nonlocal (especially, a pure entangled state), the network must be C-nontrilocal and then D-nontrilocal;

(4) The sets ${\mathrm{PT}}^{C-\mathrm{trilocal}}\left(\Omega_{3}\right)$, ${\mathrm{PT}}^{D-\mathrm{trilocal}}\left(\Omega_{3}\right)$, ${\mathrm{CT}}^{C-\mathrm{trilocal}}\left(\Delta_{3}\right)$ and ${\mathrm{CT}}^{D-\mathrm{trilocal}}\left(\Delta_{3}\right)$ are path-connectedness and have partial star-convexity.

However, the following questions are interesting and needed to be discussed further.


**Question 2.**


(Q2.1) ${\mathrm{CT}}^{C-\mathrm{trilocal}}\left(\Delta_{3}\right)={\mathrm{CT}}^{D-\mathrm{trilocal}}\left(\Delta_{3}\right)?$

(Q2.2) ${\mathrm{PT}}^{C-\mathrm{trilocal}}\left(\Omega_{3}\right)={\mathrm{PT}}^{D-\mathrm{trilocal}}\left(\Omega_{3}\right)?$


**Question 3.**


(Q3.1) $\bar{{\mathrm{CT}}^{D-\mathrm{trilocal}}\left(\Delta_{3}\right)}={\mathrm{CT}}^{D-\mathrm{trilocal}}\left(\Delta_{3}\right)?$

(Q3.2) $\bar{{\mathrm{PT}}^{D-\mathrm{trilocal}}\left(\Omega_{3}\right)}={\mathrm{PT}}^{D-\mathrm{trilocal}}\left(\Omega_{3}\right)?$


**Question 4.**


(Q4.1) $\bar{{\mathrm{CT}}^{C-\mathrm{trilocal}}\left(\Delta_{3}\right)}={\mathrm{CT}}^{C-\mathrm{trilocal}}\left(\Delta_{3}\right)?$

(Q4.2) $\bar{{\mathrm{PT}}^{C-\mathrm{trilocal}}\left(\Omega_{3}\right)}={\mathrm{PT}}^{C-\mathrm{trilocal}}\left(\Omega_{3}\right)?$

Theorem 4 implies that (Q*i*.1) and (Q*i*.2) are equivalent for each i=2,3,4.

## Figures and Tables

**Figure 2 entropy-25-00273-f002:**
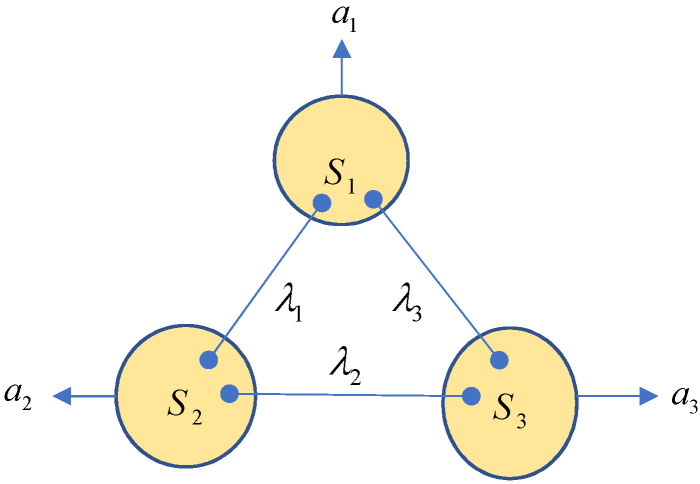
A trilocal scenario.

**Figure 3 entropy-25-00273-f003:**
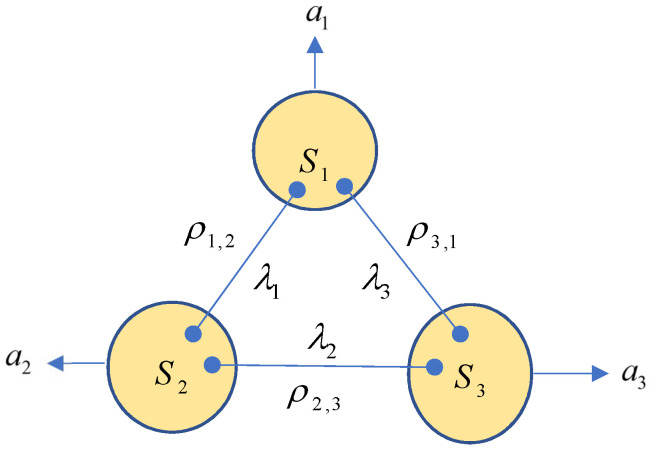
A trilocal triangle network.

## Data Availability

Not applicable.

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
