# Peer review of "Two Types of Trilocality of Probability and Correlation Tensors"

_entropy, 2023, doi:10.3390/e25020273_

Round 1

Reviewer 1 Report

The paper describes the notion of trilocality for discrete (D) or continuous (C) joint probability distributions (probability tensors) and joint conditional probability distributions (correlation tensors) in a triangle quantum network. They enumerate some formal properties of trilocal probability and correlation tensors, in particular how they can be obtained from separable or Bell local states and local POVMs. They also show that using the structure of a Euclidean vector space, then the set of trilocal correlations are path-connected and partially star-convex, where the star-convexity is with respect to three different suns, one for each marginal distribution.

I think the manuscript provides a useful clarification of what trilocality means in a triangle network scenario, a notion already  introduced by Renou, et al. In that work, Renou, et al. showed that for a particular quantum realization of a triangle network with  three bipartite entangled states shared between every pair of parties, and local POVMs for each party, they can create a probability tensor that cannot be realized by any trilocal hidden variable model (HVM). In this paper, they show that for any trilocal probability or correlation, there exists a triangle quantum network with shared separable states or Bell local states which realizes it. Intuitively, this is something one might anticipate so the result is hardly surprising but nevertheless it is nice to have the intuition made precise. 

They also demonstrate the path-connectedness and partial star-convexity of the trilocal sets, which may be theoretically interesting, but I think they probably should say what would be the physical consequences of such topological properties for more people to appreciate this. 

I think the paper as a whole presents results that are fairly straightforward to verify from the definitions so they seem sound and correct. But I am not sure these are really all that interesting. I would say the main motivation for studying network nonlocality is to have a better understanding of quantum networks. So what would really be interesting is to use what you know about trilocal correlations to make some procedure that can help identify nontrilocal ones. Maybe another helpful thing would be to use the results here to comment on some open problems on triangle networks mentioned in arXiv:2104.10700, which is a comprehensive review article on Bell nonlocality in quantum networks.

To summarize, I believe the results presented in this work are worth publishing. However I feel it need revision with more impactful results to be suitable for a journal like Entropy. 

Some questions and specific remarks:

  1. For Example 2.1 in line 112, can I redefine the probability distributions to use the absolute-values of the lambdas? If so, I believe that means I can extend it to Lambda = [-1,1]^3.

  2. For Question 2.1 in line 115, if the comment above works, does it mean that P_ball would be trilocal since it would be inscribed by the cube [-1,1]^3?

  3. On line 313-14, it should read “are path-connected and have partial star convexity.”

Author Response

Dear reviewer,
Thank you for your help. Our manuscript has been revsied according to the your and the other reviewers' reports and some changes what we made was pointed out with red-colored letters. ALso, a suggested reference [33] was added and used. I hope that this revision is satisfied with the standard of the jounal.

Thank you again.

Best wishes,
H.X. Cao

Reviewer 2 Report

Report on:

Two types of trilocality of probability and correlation tensors

by Shu Xiao, Huaixin Cao, Zhihua Guo, Kanyuan Han

Quantum networks form a natural theoretical

framework for quantum internet.

Their quantum properties demand careful investigations.

Quantum non-locality is probably the one

which attract most considerable attention due to its

fundamental role in various communication protocols.

An objective of the paper under review is to investigate properties

of the simplest yet non-trivial tri-node network.

It is known, as it is presented in the Introduction

where suitable references are quoted,

that the notion of network non-locality

may go beyond standard Bell-type scenarios.

The tri-node non-locality is the property focusing

Authors’ attention. They define and investigate

probability and correlation tensors exhibiting,

as it is shown in the work under review,

important features. The C- and D-tri-locality has

a clear physical interpretation and the results obtained

in the work are of future practical importance.

Except that the paper contains results of

more mathematical character. They are derived

in a rigorous fashion and their profs are relatively

easy to follow. Some of them are seemingly technical

but recent and older history of quantum information

development i and, in particular, quantum non-locality

shows many of the technical results sooner or later

become of practical importance. The paper is of that type:

it contains new and important theoretical concepts

which result in considerable progress of our understanding

of quantum networks. Moreover, the paper,

although demanding, is well written and the

reasoning can be followed by every patient reader.

Moreover, the list of open problems allows one

to recognize promising research directions

requiring further studies. I recommend

publication of the paper in its present form.

Author Response

(The authors gave the same response as above.)

Reviewer 3 Report

Report on ''Two types of trilocality of probability and correlation''

This manuscript focused on trilocality of probability tensors and correlation tensors based on a triangle network. The authors proposed a sufficient and necessary condition for a probability tensor that can be described by a discrete (sum) trilocal hidden variable model and a sufficient and necessary condition for a correlation tensor that can be described by a continuous (integral) trilocal hidden variable model. They further investigated the properties of various kinds of probability tensor sets.

In summary, I think these results are interesting and I recommend accepting for publication.

Author Response

(The authors gave the same response as above.)

Round 2

Reviewer 1 Report

I thank the authors for their effort in clarifying some of my questions. The point regarding Question 2.1 was particularly useful to me as it showed me some of the subtleties in thinking about trilocality, where the full product structure must be maintained over the hidden variables.

I should also say that by highlighting that Proposition 2.3 partially addresses an unresolved issue concerning the triangle network, that should be more than enough to raise some interest in these results. 

I still would have preferred a statement about some physical intuition regarding the path-connectedness. Like maybe, how unintuitive would it be if the trilocal model were not path-connected? In any case, I will let the editor decide if such clarification is still needed.

I have always felt that the technical quality of the manuscript is high. Now that it declares a partial affirmative answer to one of the open problems on quantum networks, this shows that the results are impactful. Therefore, I am happy to recommend its publication in Entropy.

Minor correction:

Line 130-131: "... the integrand may change to one that is not of the desired form."